



# A global analysis of climate-relevant aerosol properties retrieved from the network of GAW near-surface observatories

Paolo Laj[1,2,3], Alessandro Bigi[4], Clémence Rose[5], Elisabeth Andrews[6,7], Cathrine Lund Myhre[8], Martine Collaud Coen[9], Alfred Wiedensohler[10], Michael Schulz[11], John A. Ogren[7], Markus Fiebig[8], Jonas Gliβ[11], Augustin Mortier[11], Marco Pandolfi[12], Tuukka Petäjä[3], Sang-Woo Kim[13], Wenche Aas[8], Jean-Philippe Putaud[14], Olga Mayol-Bracero[15], Melita Keywood[16], Lorenzo Labrador[17], Pasi Aalto[3], Erik Ahlberg[18], Lucas Alados Arboledas[19, 20], Andrés Alastuey[12], Marcos Andrade[21], Begoña Artíñano[22], Stina Ausmeel[18], Todor Arsov[23], Eija Asmi[24], John Backman[24], Urs Baltensperger[25], Susanne Bastian[26], Olaf Bath[27], Johan Paul Beukes[28], Benjamin T. Brem[25], Nicolas Bukowiecki[29], Sébastien Conil[30], Cedric Couret[27], Derek Day[31], Wan Dayantolis[32], Anna Degorska[33], Sebastiao Martins Dos Santos[14], Konstantinos Eleftheriadis[34], Prodromos Fetfatzis[34], Olivier Favez[35], Harald Flentje[36], Maria I. Gini[34], Asta Gregorič[37], Martin Gysel-Beer[25], Gannet A. Hallar[38], Jenny Hand[31], Andras Hoffer[39], Christoph Hueglin[40], Rakesh K. Hooda[24,41], Antti Hyvärinen[24], Ivo Kalapov[23], Nikos Kalivitis[42], Anne Kasper-Giebl[43], Jeong Eun Kim[44], Giorgos Kouvarakis[42], Irena Kranjc[45], Radovan Krejci[46], Markku Kulmala[3], Casper Labuschagne[47], Hae-Jung Lee[44,*], Heikki Lihavainen[24], Neng-Huei Lin[48], Gunter Löschau[26], Krista Luoma[3], Angela Marinoni[2], Frank Meinhardt[27], Maik Merkel[10], Jean-Marc Metzger[49], Nikolaos Mihalopoulos[42,50], Nhat Anh Nguyen[51], Jakub Ondracek[52], Noemi Pérez[12], Maria Rita Perrone[53], Jean-Eudes Petit[54], David Picard[5], Jean-Marc Pichon[5], Veronique Pont[55], Natalia Prats[21], Anthony Prenni[56], Fabienne Reisen[16], Salvatore Romano[53], Karine Sellegri[5], Sangeeta Sharma[57], Gerhard Schauer[43], Patrick Sheridan[7], James Patrick Sherman[58], Maik Schütze[27], Andreas Schwerin[27], Ralf Sohmer[27], Mar Sorribas[59], Martin Steinbacher[40], Junying Sun[60], Gloria Titos[12,19,20], Barbara Toczko[61], Thomas Tuch[10], Pierre Tulet[62], Peter Tunved[46], Ville Vakkari[24], Fernando Velarde[21], Patricio Velasquez[63], Paolo Villani[5], Sterios Vratolis[34], Sheng-Hsiang Wang[48], Kay Weinhold[10], Rolf Weller[64], Margarita Yela[59], Jesus Yus-Diez[12], Vladimir Zdimal[52], Paul Zieger[46], Nadezda Zikova[52],

[1]Univ. Grenoble-Alpes, CNRS, IRD, Grenoble-INP, IGE, 38000 Grenoble, France
[2]Institute for Atmospheric and Earth System Research, University of Helsinki, Helsinki, Finland
[3] Institute of Atmospheric Sciences and Climate, National Research Council of Italy, Bologna, Italy
[4] Università di Modena e Reggio Emilia, Department of Engineering "Enzo Ferrari", Modena, Italy
[5]Université Clermont-Auvergne, CNRS, LaMP, OPGC, Clermont-Ferrand, France
[6]Cooperative Institute for Research in Environmental Sciences, University of Colorado, Boulder, Colorado, USA
[7]NOAA/Earth Systems Research Laboratory, Boulder, CO, USA
[8]NILU, Norwegian Institute for Air Research, Kjeller, Norway
[9]Federal Office of Meteorology and Climatology, MeteoSwiss, Payerne, Switzerland
[10]Institute for Tropospheric Research, Leipzig, Germany
[11] Norwegian Meteorological Institute, Oslo, Norway
[12] Institute of Environmental Assessment and Water Research (IDAEA), Spanish Research Council (CSIC), Barcelona, Spain
[13] School of Earth and Environmental Sciences, Seoul National University, Seoul, Korea
[14] European Commission, Joint Research Centre (JRC), Ispra, Italy
[15]University of Puerto Rico, Rio Piedras Campus, San Juan, Puerto Rico


[16] CSIRO Oceans and Atmosphere, PMB1 Aspendale VIC, Australia

[17] World Meteorological Organisation, Global Atmosphere Watch Secretariat, Geneva, Switzerland

[18] Lund University, Department of Physics, Division of Nuclear Physics, P.O. Box 118, 221 00 Lund, Sweden

[19] Department of Applied Physics, University of Granada, Granada, Spain

[20] Andalusian Institute for Earth System Research (IISTA-CEAMA), University of Granada, Autonomous Government of Andalusia, Granada, Spain

[21] Universidad Mayor de San Andres, Laboratorio de Fisica de la Atmosfera, La Paz, Bolivia

[22] Izaña Atmospheric Research Center (IARC), State Meteorological Agency (AEMET), Santa Cruz de Tenerife, Spain

[23] Institute for Nuclear Research and Nuclear Energy, Bulgarian Academy of Sciences, Sofia, Bulgaria

[24] Atmospheric composition research, Finnish Meteorological Institute, Helsinki, Finland

[25] Laboratory of Atmospheric Chemistry, Paul Scherrer Institute, Villigen PSI, Switzerland

[26] Saxon State Office for Environment, Agriculture and Geology (LfULG), Dresden, Germany

[27] German Environment Agency (UBA), Zugspitze, Germany.

[28] Unit for Environmental Sciences and Management, North-West University, Potchefstroom, ZA-2520, South Africa

[29] Atmospheric Sciences, Department of Environmental Sciences, University of Basel, Basel, Switzerland

[30] ANDRA DRD/GES Observatoire Pérenne de l'Environnement, 55290 Bure, France

[31] Cooperative Institute for Research in the Atmosphere, Colorado State University/ National Park Service

[32] Meteorological Climatological and Geophysical Agency (BMKG), Jakarta, Indonesia

[33] Institute of Environmental Protection – National Research Institute, Warsaw, Poland

[34] Institute of Nuclear and Radiological Science & Technology, Energy & Safety N.C.S.R. "Demokritos", Attiki, Greece

[35] Institut National de l'Environnement Industriel et des Risques (INERIS), Verneuil-en-Halatte, France

[36] German Weather Service, Meteorological Observatory Hohenpeissenberg, Hohenpeißenberg, Germany

[37] Aerosol d.o.o., Ljubljana, SI-1000, Slovenia

[38] Department of Atmospheric Sciences, University of Utah, Salt Lake City, UT 84112

[39] MTA-PE Air Chemistry Research Group, Veszprém, Hungary

[40] Empa, Swiss Federal Laboratories for Materials Science and Technology, Duebendorf, Switzerland

[41] The Energy and Resources Institute, IHC, Lodhi Road, New Delhi, India

[42] Environmental Chemical Processes Laboratory (ECPL), University of Crete, Heraklion, Crete, 71003, Greece

[43] ZAMG - Sonnblick Observatory Freisaalweg 165020 Salzburg Austria

[44] Environmental Meteorology Research Division, National Institute of Meteorological Sciences, Seogwipo, Korea

[45] Hydrometeorological Institute of Slovenia, Ljubljana, Slovenia

[46] Department of Environmental Science and Analytical Chemistry (ACES) & Bolin Centre for Climate Research, Stockholm University, S-10691 Stockholm, Sweden

[47] South African Weather Service, Research Department, Stellenbosch, South Africa.

[48] Department of Atmospheric Sciences, National Central University, Taoyuan, Taiwan

[49] Observatoire des Sciences de l'Univers de La Réunion (OSUR), UMS3365, Saint-Denis de la Réunion, France

[50] Institute of Environmental Research & Sustainable Development, National Observatory of Athens, Palea Penteli, 15236, Greece

[51] Hydro-Meteorological Observation Center (HYMOC), Vietnam Meteorological and Hydrological Administration (VNMHA), Ministry of Natural Resources and Environment (MONRE), Ha Noi, Vietnam

[52] Department of Aerosol Chemistry and Physics, Institute of Chemical Process Fundamentals, CAS, Prague, Czech Republic

[53] Consorzio Nazionale Interuniversitario per le Scienze Fisiche della Materia and Università del Salento, Lecce, Italy

[54] Laboratoire des Sciences du Climat et de l'Environnement, LSCE/IPSL, UMR 8212 CEA-CNRS-UVSQ, Université Paris-Saclay, Gif-sur-Yvette, France

[55] Laboratoire d'Aérologie, CNRS-Université de Toulouse, CNRS, UPS, Toulouse, France

[56] National Park Service, Air Resources Division, Lakewood, CO, USA

[57] Environment and Climate Change Canada, Toronto, ON, Canada

[58] Department of Physics and Astronomy, Appalachian State University, Boone, NC USA

[59] Atmospheric Sounding Station, El Arenosillo, Atmospheric Research and Instrumentation Branch, INTA, 21130, Mazagón, Huelva, Spain



[60]State Key Laboratory of Severe Weather & Key Laboratory of Atmospheric Chemistry of CMA, Chinese Academy of Meteorological Sciences, Beijing 100081, China

[61]Department of Environmental Monitoring, Assessment and Outlook, Chief Inspectorate of Environmental Protection, Warsaw, Poland

[62]Laboratoire de l'Atmosphère et des Cyclones (LACy), UMR8105, Université de la Réunion – CNRS – Météo-France, Saint-Denis de La Réunion, France

[63]Climate and Environmental Physics, University of Bern, Bern, Switzerland

[64]Alfred Wegener Institute, 27570 Bremerhaven, Germany

* Now at National Council on Climate and Air Quality, Seoul, Korea

*Correspondence to*: Paolo Laj (paolo.laj@univ-grenoble-alpes.fr)

**Abstract.** Aerosol particles are essential constituents of the Earth's atmosphere, impacting the earth radiation balance

directly by scattering and absorbing solar radiation, and indirectly by acting as cloud condensation nuclei. In contrast to most greenhouse gases, aerosol particles have short atmospheric residence time resulting in a highly heterogeneous distribution in space and time. There is a clear need to document this variability at regional scale through observations involving, in particular, the in-situ near-surface segment of the atmospheric observations system. This paper will provide the widest effort so far to document variability of climate-relevant in-situ aerosol properties (namely wavelength dependent particle light

scattering and absorption coefficients, particle number concentration and particle number size distribution) from all sites connected to the Global Atmosphere Watch network. High quality data from almost 90 stations worldwide have been collected and controlled for quality and are reported for a reference year in 2017, providing a very extended and robust view of the variability of these variables worldwide. The range of variability observed worldwide for light scattering and absorption coefficients, single scattering albedo and particle number concentration are presented together with preliminary

information on their long-term trends and comparison with model simulation for the different stations. The scope of the present paper is also to provide the necessary suite of information including data provision procedures, quality control and analysis, data policy and usage of the ground-based aerosol measurements network. It delivers to users of the World Data Centre on Aerosol, the required confidence in data products in the form of a fully-characterized value chain, including uncertainty estimation and requirements for contributing to the global climate monitoring system.

**1 Introduction**

Climate change is perceived as one of the world's greatest threats with the potential to undermine the three social, economic and environmental pillars of sustainability. Changing atmospheric composition is one of the important drivers of climate change acting both on the global scale (i.e. warming related to long-lived greenhouse gases such as $CO_2$) and on the regional scale where atmospheric compounds with shorter lifetime may enhance or slightly reduce warming from long-lived

greenhouse gases.





Aerosol particles are essential constituents of the Earth's atmosphere, impacting the earth's radiation balance directly by scattering and absorbing solar radiation, and indirectly by acting as cloud condensation nuclei. In the recent IPCC Reports on Climate Change (AR5), the impact of aerosols on the atmosphere is widely acknowledged as still one of the most significant

and uncertain aspects of climate change projections (IPCC, 2013, Bond et al., 2013). The magnitude of aerosol forcing is estimated to be –0.45 (–0.95 to +0.05) W m$^{-2}$ for aerosol alone and –0.9 (–1.9 to –0.1) W m$^{-2}$ when aerosol/cloud feedbacks are accounted for, both with medium confidence level. A more recent study by Lund et al (2018) report aerosol direct radiative forcing of −0.17 W m$^{-2}$ for the period 1750 to 2014, significantly weaker than the IPCC AR5 2011–1750 estimate. Differences are due to several factors, including stronger absorption by organic aerosol, updated parameterization of Black

Carbon (BC) absorption in the applied model, and reduced sulphate cooling.

The mechanisms by which aerosol particles influence the Earth's climate have been subject to numerous studies in the last decades and are well understood, yet the uncertainty of the anthropogenic forcing still remains the largest uncertainty among the factors influencing changes in climate. In contrast to most greenhouse gases, aerosol particles have short atmospheric

residence time (days) and undergo transport, mixing, chemical aging, and removal by dry and wet deposition, resulting in a highly heterogeneous distribution in space and time. Different parameterizations used to calculate atmospheric mass loads lead to high diversity among global climate models (Textor et al., 2006; Huneeus et al., 2011; Tsigaridis et al., 2014; Bian et al., 2017). There are several reasons for the high uncertainty: uncertainties associated with aerosol and aerosol precursor emissions linked to new particle formation, in particular for the pre-industrial period; uncertainties in the representation of

the climate-relevant properties of aerosol including the representation of the pre-industrial conditions); uncertainties in the parametrization of sub-grid processes in climate models, in particular for cloud processes (updraft velocity, cloud liquid water content, cloud fraction; relationship between effective radius and volume mean radius, impact of absorbing impurities in cloud drop single scattering albedo, etc.); and uncertainties in providing an adequate characterization of aerosol climate-relevant properties (spatial and temporal variability). A study published by Carslaw and coworkers (2013) has shown that

45% of the variance of aerosol forcing arises from uncertainties in natural precursor emissions, also in line with the results of Lund et al. (2018).

The study of Lund et al. (2018) also highlights the importance of capturing regional emissions and verification with measurements. Natural and anthropogenic emissions of primary aerosol and their gaseous precursors have been estimated at

different scales in many studies and inventories are now providing fairly accurate information on historical emission trends. Historical emission estimates for anthropogenic aerosol and precursor compounds are key data needed for assessing aerosol impact on climate, but are difficult to obtain with precision and there are important discrepancies amongst different estimates even for key aerosol climate forcers like black carbon (Granier et al., 2011; Klimont et al., 2017; Lamarque et al., 2010; Wang et al., 2014). For example, in a recent study using ice-core records from Alpine regions, Lim et al. (2017) showed that

BC emission inventories for the period 1960s–1970s may be strongly underestimating European anthropogenic emissions.





Providing reliable observations of aerosol properties relevant to climate studies at spatial and temporal resolution suited to users is essential. For example, a measured decrease in pollutant concentrations would be the ultimate indicator of a successful policy to reduce emissions. However this requires long-term production and delivery of science-based data of
known quality in terms of precision, accuracy and sufficient density of data points over the region of interest for the measurements to be representative. Similarly, evaluating model performances from comparisons with observations requires that sets of high quality data are made available in comparable formats, with known uncertainties so that comparisons are meaningful. Current modelling tools are suited to the diversity of applications required by the disparate spatial and temporal scales of atmospheric impacts on climate, human health and ecosystems. There is still a need for accurate representation of
observed aerosol which remains challenging, leading to considerable diversity in the abundance and distribution of aerosols among global models. Capacity exists to deliver information products in a form adapted to climate policy applications in particular, but models need to be validated against measured atmospheric composition both in the short- and long-term (Benedetti et al., 2018).

One major aspect of aerosol forcing on climate is linked to its multi-variable dimension: optical properties of an aerosol particle population are closely linked to its chemical, physical and hygroscopic properties and also to the altitude-dependency of these parameters, which undergo significant short-term (diurnal) temporal variations. The effects of aerosol on climate are driven by both extensive and intensive aerosol properties. Aerosol extensive properties depend on both the nature of the aerosol and the aerosol particle concentration. In contrast, intensive properties are independent of particle
concentration and instead relate to intrinsic properties of the aerosol particles (Ogren, 1995). Table 1 lists properties relevant to the determination of aerosol climate forcing. We use the terminology proposed by OSCAR (https://www.wmo-sat.info/oscar/) and Petzold et al. (2013) for the specific case of black carbon. Some of the aerosol properties in Table 1 are recognized as aerosol Essential Climate Variables (ECVs) products for climate monitoring in the Global Climate Observing System (GCOS). The WMO/GAW Report No. 227 (2016) provides a synthesis of methodologies and procedures for
measuring the recommended aerosol variables within the GAW network. The report identifies a list of comprehensive aerosol measurements to be conducted as a priority as well as core measurements to be made at a larger number of stations.
It is clear that neither a single approach to observing the atmospheric aerosol nor a limited set of instruments can provide the data required to quantify aerosol forcing on climate in all its relevant dimensions and spatial/temporal scales (Kahn et al., 2017; Anderson et al., 2005). Observations from space through remote sensing methods are providing key information to
accurately document extensive properties but are still not sufficient to provide information with the required degree of spatial and temporal resolution needed for many applications. Further, remote sensing retrievals have only limited capabilities for determining aerosol chemistry, aerosol particle light absorption, particle size number distribution, Condensation Nuclei (CN), Cloud Condensation Nuclei (CCN) and Ice Nuclei (IN) (Kahn et al., 2017). Instead, in situ observations from stationary surface observatories, ships, balloons, and aircraft provide very detailed characterizations of the atmospheric



aerosol, often on limited spatial scales. Non-continuous mobile platforms such as aircraft and balloons provide the vertical dimension, however, with limited temporal resolution. The current availability and accessibility of ground-based datasets on climate relevant aerosol properties vary substantially from place to place. An aerosol observing system for climate requires that all the types of observations are combined with models to extrapolate measurement points to large geographical scales against which satellite measurements can be compared (e.g., Anderson et al., 2005, Petäjä et al., 2016).


The in-situ segment of atmospheric observations is very complex and involves multiple partners, some are organized in measurement networks, active at regional or global scales, some are working almost independently. Networks support consistent, long-term measurements of atmospheric variables in order to detect trends and assess reasons for those trends. Information on the variability of aerosol properties from ground-based stations can mainly be divided into two types: (i) in-

situ networks driven by policy initiatives, with a relatively close relationship to stakeholders and often structured at country scale, providing limited sets of aerosol variables and (ii) the research-based networks, organized at continental or international scales particularly focusing on climate-relevant parameters. The Global Atmosphere Watch (GAW) Programme of the World Meteorological Organization (WMO) was established in 1989 and the GAW aerosol measurement programme in 1997 originally dedicated to monitoring of climate-relevant species. Networks contributing to the provision of climate

relevant aerosol properties are mainly structured with three different categories, some of them affiliated to GAW as contributing networks and some other operating independently:

- Networks for the detection of Aerosol Optical Depth (AOD): AERONET (https://aeronet.gsfc.nasa.gov/), GAW PFR (http://www.pmodwrc.ch/worcc/) and CARSNET (China Aerosol Remote Sensing NETwork, Che et al.,
2009). Aerosol optical depth (AOD) is one of five core aerosol variables recommended for long- term continuous measurements in the GAW programme.
- Networks for the detection of aerosol profiles which are internationally organized into GALION (GAW Aerosol LIdar Observing Network) and composed of lidar instruments operating within NDACC (Network for the Detection of Atmospheric Composition Changes), EARLINET/ACTRIS (European Atmospheric Lidar Network) and
MPLNET, principally ADNET in Asia and MPLNET. Other lidars (CLN, CORALNET, ALINE) contribute to GALION goals but are not at the same level of maturity or are solely regional in extent.
- Networks for the detection of in-situ aerosol properties, mainly divided into contributions from NOAA's Federated Aerosol Network (NFAN), encompassing sites primarily in North America but also including sites in Europe, Asia, and the southern hemisphere, including Antarctic sites (NFAN, Andrews et al., 2019) and ACTRIS
(https://actris.eu) in Europe, but also including sites other WMO regions (https://cpdb.wmo.int/regions). In Europe, the European Monitoring and Evaluation Programme' EMEP (https://www.emep.int), and, in the US, the IMPROVE network (http://vista.cira.colostate.edu/Improve/) are also providing key information on aerosol in-situ variables (Tørseth et al, 2012). Additional networks contributing to the provision of in-situ aerosol properties are





the Canadian Air and Precipitation Monitoring Network (CAPMoN), the Acid Deposition Monitoring Network in
East Asia (EANET) and the Korea Air Quality Network (KRAQNb)

Finally specific contributions are brought by the vertical profiles to in-situ observations routinely performed by IAGOS (In-flight Atmospheric Observing System), a contributing network to GAW and by additional ground-based observations operated outside the GAW context, such as SPARTAN (https://www.spartan-network.org).

**2 Scope of the paper**

The scope of the present paper is to provide the necessary suite of information to define a fully traceable ground-based aerosol measurements network, and to give an overview of the state of the operation in the network for a reference year. The paper should deliver to users of the World Data Centre on Aerosol (WDCA), the required confidence in data products in the
form of a fully-characterized value chain, including uncertainty estimation and requirements for climate monitoring.

The paper is limited to a subset of the climate-relevant aerosol variables. It focuses on variables that are measured or derived from near-surface measurements, thus excluding all columnar and profile variables, despite their strong climate relevance. A second criteria for discussion in the paper is connected to the fact that long-term information is available at sufficient sites
across the globe to derive trends and variability with sufficient robustness. Clearly, for many of the variables listed in Table 1, information is only available from a number of stations that are either almost exclusively documenting one single region (i.e. measurements of aerosol chemical properties with online aerosol mass spectrometers in Europe only) or not numerous enough to provide a robust assessment. In the case of EC/OC observations for example, information exists for many sites in different WMO regions but many of them no longer documented at the WDCA.

Finally, the last criteria is connected to the quality, intercomparability and accessibility of measurements worldwide, meaning that all information used in the paper must be well documented with rich metadata, traceable in provenance and quality, and accessible for all. This clearly limits the scope of the paper to the four independent climate-relevant variables mentioned above: i) particle light scattering coefficient, ii) particle light absorption coefficient, iii) particle number
concentration, and iv) particle number size distribution.

For this set of variables, there has been, in the last decades, a significant international effort to harmonize the practice and methodologies across the frameworks, and strengthen systematic observations through different networks, or research infrastructure in the case of Europe, operating with a certain degree of interoperability. All networks jointly defined standard
operation procedures (SOPs), conduct data collection in a timely and systematic manner, and promote open access and





exchange of data without restriction through a unique data hub, the WDCA, hosted by NILU in Norway (https://www.gaw-wdca.org/). Operators from these networks perform joint assessments and analyses of data resulting in scientific publications that are discussed below.

This paper then provides a full-characterization of the value chain for these four aerosol variables that will serve for defining the fiducial reference network in the future. It also provides an overview of the variability of the variables, and of some additional derived variables from the collection of data for the reference year 2017. The present paper is jointly written with companion papers, three of which one (Collaud Coen et al., (submitted), Gliβ et al., (submitted) and Mortier et al., submitted) are submitted in parallel with this paper. Gliβ et al., (submitted) and Mortier et al., (submitted) also belong to the

AeroCom initiative for IPCC. Papers are the following:

- Collaud Coen et al. (submitted) analyses trends and variability of SARGAN optical properties using continuous observations worldwide
- Gliβ et al. (submitted) uses the AeroCom (Aerosol Comparisons between Observations and Models, https://aerocom.met.no/) models to assess performances of global-scale model performance for global and regional
SARGAN variables distributions, and variability,
- Mortier et al. (submitted) is a multi-parameter analysis of the aerosol trends over the last two decades comparing the output from AEROCOM models and observations, including time series of SARGAN aerosol optical variables.
- Additional papers are in preparation to analyse the variability of SARGAN physical properties and to investigate the variability of carbonaceous aerosol using continuous observations worldwide

Some preliminary information on trends and comparisons with models that are further developed in Collaud Coen et al. (submitted), Gliβ et al. (submitted) and Mortier et al. (submitted) are presented in this paper. Additional manuscript are in preparation to further investigate variability of the optical and physical properties.

This paper is integrated into a larger initiative called SARGAN (in-Situ AeRosol GAW Network) that will serve as the
equivalent for GALION for the near-surface observations of aerosol variables. It is intended to support a future application of SARGAN, and possibly other components of the GAW network, to become a GCOS associated network (https://gcos.wmo.int/en/networks). This requires the definition of threshold, breakthrough and goals for spatial and temporal resolutions that may be used for designing an operational aerosol in-situ network suited to global monitoring requirements in GCOS. Finally, this paper documents all elements required for establishing the GCOS network by addressing 1) the
procedures for collecting and harmonizing measurements, data, metadata and quality control, 2) procedures for curation and access to SARGAN data, 3) the available harmonised surface observations within SARGAN and status of the station network, 5) the present-day distribution of SARGAN aerosol properties and 5) requirements for using SARGAN for global climate monitoring applications.





**3 Procedures for collecting and harmonizing measurements, quality control, and data curation and access**

Controlling and improving data quality and enhancing their use by the scientific community is an essential aim within observational networks. Procedures are continuously evolving as new instruments become commercially available and because efforts from the scientific community have resulted in more appropriate operation procedures for monitoring purposes. In the last decade, significant progress has been made in the harmonization of measurement protocols across the different networks and to ensure that all information is made readily available in a coordinated manner


In the GAW program, the individual station and its host organization are scientifically responsible for conducting the observations according to the standard operating procedures. This responsibility includes quality assurance of the instruments, as well as quality control of the data after measurement. In quality assurance, the stations collaborate with dedicated calibration centers, usually by sending their instruments for off-site calibration in regular intervals, and by station

audits performed by relevant GAW Calibration centers

**3.1 Harmonization of measurement protocols in SARGAN**

Improving data quality and enhancing data use by the scientific community is an essential aim within GAW and the contributing networks. The measurement guidelines and standard operating procedures (SOPs) used for aerosol in situ measurements within GAW are discussed and prepared by Scientific Advisory Group (SAG) on "Aerosol" and accepted by

the scientific community through peer-reviewed processes. The SOPs provide guidelines for good measurement practice and are listed in WMO/GAW report #227 (2016) and connected reports.

The knowledge of the aerosol effect on climate and air quality as well as the techniques used for the determination of the essential aerosol variables to be monitored at ground-based sites have evolved considerably in the last decade. The

methodologies, guidelines and SOPs are often elaborated and tested within the regional networks such as NFAN or the European research infrastructure ACTRIS, and transferred to the GAW program to be adopted as Guidelines or more operational SOPs. SOPs are now available for almost all aerosol climate-relevant measurements, including for some of the most recent aerosol instruments.

The general guidelines for in-situ aerosol measurements in GAW are given in the general WMO/GAW report #227 (2016) and in specific GAW reports such as WMO/GAW Report #200 (2011) for particle light scattering and absorption coefficients. Some of the recommended procedures are also adopted at a level of recommended standards by other bodies, such as EMEP under the UNECE, CEN (Center for European Normalization). This is the case for the measurement of the particle number concentration with condensation particle counters (CEN/TS 16976) as well as for the particle number size

distribution with mobility particle size spectrometers (CEN/TS 17434).



In SARGAN, measurements of the particle light scattering coefficient are performed using integrating nephelometers, while measurements of the particle light absorption coefficient utilize various filtered-based absorption photometer instruments. Both particle light scattering and absorption coefficients are dependent upon the size, shape, and composition of the particles as well as the wavelength of the incident light. Measurements of the particle light scattering and absorption coefficients ideally would be performed at various wavelengths at a defined relative humidity. In GAW and the contributing networks, in-situ microphysical and optical aerosol measurements should be performed for a relative humidity (RH) lower than 40%, although some stations allow measurements up to 50%.

Furthermore, information on the relative amounts of particle light scattering vs. absorption is required for radiative forcing calculations and is defined by the aerosol single scattering albedo, $\omega_0$, which is the ratio of the particle light scattering coefficient over the particle light extinction coefficient, as defined in Table 1: $\omega_0 = \sigma_{sp}/(\sigma_{sp} + \sigma_{ap})$. In this article, $\omega_0$ is computed for one specific $\lambda$ (550 nm). The scattering Ångström exponent, AE, defined by the power-law $\sigma_{sp} \propto C_0 \lambda^{-AE}$, describes the wavelength-dependence for scattered light and is an indicator of particle number size distribution, and, thus, on the type of aerosol such as anthropogenic, mineral dust or sea salt. The scattering Ångström exponent can be directly derived from the measured particle light scattering coefficients at different wavelengths.

Müller et al. (2011) performed an intercomparison exercise for integrating nephelometers to propose procedures for correcting the non-ideal illumination due to truncation of the sensing volumes in the near-forward and near-backward angular ranges and for non-Lambertian illumination from the light sources. Müller's work expanded the initial findings of Anderson and Ogren (1998), which were for a specific nephelometer model. Additionally, measurements of the dependence of the particle light scattering coefficient on the relative humidity are essential for the calculation of aerosol radiative effects in the atmosphere. This enhanced particle light scattering due to water take-up is strongly dependent on the particle number size distribution and the size-resolved particle composition. However, such measurements require an additional instrumental set-up, which has been implemented at only at very few stations and, with few exceptions, only on a campaign basis (Burgos et al., 2019; Titos et al., 2016).

Petzold and Schönlinner (2004) developed the filter-based Multi-Angle Absorption Photometer (MAAP), which can determine the particle light absorption coefficient directly, considering the light attenuation through and the backscattering above the filter. For other filter-based absorption photometers, the particle light absorption coefficient is determined from the light attenuation through the filter, considering scattering cross-sensitivities and loading effects. The procedures to correct for scattering cross-sensitivity in Particle Soot Absorption Photometer (PSAP) instruments are described in Bond et al. (1999) and Ogren (2010). Several correction procedures for Aethalometers are given in Collaud Coen et al. (2010). Recently,





the ACTRIS community developed a harmonized factor for the AE31 to determine the particle light absorption coefficient,
based on long-term intercomparison between Aethalometers and the MAAP for different environments and aerosol types
(WMO/GAW report #227, 2016).

The physical aerosol particle properties reported in this article are derived from the particle number concentration and
number size distribution limited to the ultrafine and fine range. These measurements are performed using condensation
particle counters (CPC) and mobility particle size spectrometers (MPSS). Wiedensohler et al. (2012) describes procedures
for long-term MPSS measurements and for their quality assurance. Since measurements of particle number size distributions
are mainly restricted to ACTRIS sites and at a few other stations, a global assessment on aerosol physical properties can be
only derived for the particle number concentration. For sites, where only MPSS data are available, the particle number
concentration is determined from the integral over the particle number size distribution measured by the MPSS (see section
5.2 for discussion).  Table 2 below summarizes all technical information related to the measurements of aerosol optical and
physical properties in SARGAN.

### 3.2 Curation and access to SARGAN data

In the management of data throughout its lifecycle, data curation is the activity that collects, annotates, verifies, archives,
publishes, presents, and ensures access to all persistent data sets produced within the measurement framework and program.
The main purpose of data curation is to ensure that data are reliable and accessible for future research purposes and reuse. To
this end, SARGAN data should be traceable to the original raw observational data, include version control and identification
in case of updates, and include rich metadata going beyond discovery metadata (e.g., variable and station information) to use
metadata (instrument description, operating procedures, station setting, calibration and quality assurance measures and
uncertainties). SARGAN data are archived at WDCA, which is the data repository for microphysical, optical, and chemical
properties of atmospheric aerosol for the WMO/GAW programme.

To ensure traceability of data products, WDCA uses a system of 3 data levels:
- Level 0: annotated raw data, all parameters provided by instrument, parameters needed for further processing,
format is instrument model specific format, "native" time resolution.
- Level 1: data processed to final parameter, calibrations applied, invalid and calibration episodes removed, format is
property specific, "native" time resolution, conversion to reference conditions of temperature and pressure (273.15
K, 1013.25 hPa).
- Level 2: data aggregated to hourly averages, atmospheric variability quantified, format is property specific.




Each higher data level is produced from the respective lower level as specified by the pertaining operating procedure. The templates for data level and instrument are published on the WDCA homepage and pages referenced from there, together with references to the relevant operating procedures. The templates indicate the metadata and data elements (discovery and use metadata) expected when submitting data to WDCA, which have been specified in collaboration with the GAW

scientific advisory group (SAG) for aerosol and the GAW World Calibration Centre for Aerosol Physics (WCCAP) to ensure that relevant and useful metadata are collected.

Stations report data to WDCA on an annual basis. After quality control, the station submits the data to WDCA via an online, web-based submission tool: https://ebas-submit-tool.nilu.no. In this process, the tool gives immediate feedback on syntax

errors, and performs checks on semantics and sanity of both metadata and data. During curation at WDCA, the data files are inspected both automatically and manually for metadata completeness and consistency, while the data are inspected for outliers, spikes, and sanity. Issues discovered in the process are reported back to the station, and the station asked to take corrective action and resubmit the data. The same applies for issues discovered after data publication.

By joining the GAW programme, stations commit to reporting their observations in a fully and manually quality controlled version (level 2) on an annual basis, with a deadline of 31 December of the year following the data year to be reported. WDCA encourages stations to report their data in a traceable way, i.e. to include data level 0 and 1 with their submissions.

GAW guidelines for quality control have developed and improved over the lifetime of the programme. At the beginning,

quality control reflected the GAW objective of providing observations of atmospheric compositions with large scale representativity. For this reason, observations influenced by local and regional emissions, or by regional phenomena, were flagged invalid during quality control and excluded from being archived. Later, it was acknowledged that atmospheric composition data serves multiple purposes and applications. This is reflected by the recommendation to only remove data affected by instrument issues or contamination during quality control, and indicate local or regional influence with a flag that

leaves the data valid. This implies, for any application of WDCA data, filtering the data according to purpose is the first step. When using WDCA data, this shift in quality control approach, which may vary among stations due to their scientific independence, needs to be taken into account.

The Global Atmosphere Watch, and the affiliated networks have agreed on a FAIR-use data policy encouraging an unlimited

and open data policy for non-commercial use, provided without charge, unless noted otherwise. Users of WDCA are encouraged to contact and eventually offer co-authorship, to the data providers or owners whenever substantial use is made of their data. Alternatively, acknowledgement must be made to the data providers or owners and to the project name when these data are used within a publication. All data related to the present article are available at the WDCA.



**4 Procedures for collecting and harmonizing measurements, quality control, and data curation and access**

**4.1 A short history of aerosol monitoring networks**

The first network designed to make long-term measurements of climate-relevant aerosol properties was the Geophysical Monitoring for Climate Change (GMCC) program, formed by NOAA in the early 1970's. GMCC was "designed to establish and maintain a program of observation and analysis of data representative of the global background of selected gases and aerosols" This focus on establishing a global background climatology meant that the stations were located at remote sites, far

from human emission sources, in order to ascertain the extent to which human activities caused changes in climate-relevant aerosol properties. The four initial GMCC stations were chosen to sample representative latitudes within both hemispheres - polar, mid-latitude, and tropical, and were located at South Pole, Antarctica; Point Barrow, Alaska; Mauna Loa, Hawaii; and Cape Matatula, American Samoa. Two additional locations were initially planned, on the west coast of the USA and on or eastward of the east coast of the USA, but were not established until much later. As a consequence of the site selection

criteria, the GMCC stations were not positioned to characterize the climate-forcing properties of aerosols in the regions where the climate forcing was large, a weakness that was not addressed until the 1990's when NOAA established stations in and downwind of the continental USA and the GAW network was founded.

Aerosol particle number concentration was the first aerosol property measured at the GMCC stations, initially with manual

expansion-type, water-based instruments and later with automated versions. The rationale for the choice of this variable was that these very small particles "are present in all forms of combustion [products], such as those from automobiles, coal or oil-burning power plants, and other human activities, it is essential to monitor the background tropospheric aerosol concentration in order to assess man's possible impact on his global environment". Recognizing that aerosols may play an important role in the global radiation balance, because they influence the heat budget and scatter or absorb both incoming

solar radiation and outgoing terrestrial radiation, multi-wavelength measurements of aerosol particle light scattering coefficient using integrating nephelometers were added at the four GMCC stations in the mid- to late-1970's.

Although measurements of aerosol particle number concentration and light scattering coefficient were made during multiple, short-term field studies and in long-term studies at individual field stations (e.g., Gras, 1995), the next network to be

established for these measurements was the IMPROVE (Interagency Monitoring of Protected Visual Environments) network in the USA, which was initiated in 1985 to monitor visibility degradation in US National Parks and Wilderness Areas. Nephelometer data from 12 IMPROVE sites, most beginning in 1993, were included in the Collaud Coen et al. (2013) trend analysis.





After the establishment of the WMO GAW program in 1989, a meeting of experts was convened in 1991 to consider the aerosol component of GAW (GAW Report #79). This group formulated the objective of the GAW aerosol program to understand changes in the atmospheric aerosol, with two specific tasks:

    a) to assess the direct and indirect effect of aerosol on climate - through aerosol data representative of different regions; and

b) to determine the relative contribution of natural and man-made sources to the physical and chemical properties of the aerosol at locations representative of different regions.

The objective of the GAW aerosol program was reformulated at the first meeting of the GAW Scientific Advisory Group (SAG) for Aerosols in 1997 to determine the spatio-temporal distribution of aerosol properties related to climate forcing and air quality up to multidecadal time scales and further refined in WMO/GAW Report #153 (2003) to determine the spatio-

temporal distribution of aerosol properties related to climate forcing and air quality on multi-decadal time scales and on regional, hemispheric and global spatial scales.

Under the leadership of SAG-Aerosols, the GAW aerosol network grew slowly through the decade 1997-2007, with the refinement of recommended measurements and sampling procedures (WMO/GAW Report #153, 2003), and the

establishment of the World Data Center for Aerosols (WDCA) and the World Calibration Center for Aerosol Physical Properties (WCAAP). The GAW aerosol network was greatly strengthened, particularly in Europe, by the establishment of the EUSAAR (European Supersites for Atmospheric Aerosol Research) program in 2006 and its successor ACTRIS (Aerosols, Clouds and Trace gases Research Infrastructure) in 2011. The expansion of the GAW aerosol network was further enhanced by the NOAA Federated Aerosol Network (Andrews et al., 2019), which currently supports nearly 30 GAW

aerosol stations with scientific and technical advice, data acquisition software, and streamlined procedures for submitting quality-controlled data to WDCA.

### 4.2 An overview of recent studies of variability and trends of aerosol in-situ optical and physical properties

The pioneering works of Bodhaine (1983; 1995), Delene and Ogren (2002) for US sites, and Putaud et al. (2004 and 2010), and Van Dingenen et al. (2004) for European sites are the first studies documenting variability of climate-relevant aerosol

properties using long term observations performed at the network scale. Using long term observations performed at several sites across the US, Delene and Ogren (2002) investigated the systematic relationships between aerosol optical properties and aerosol loadings that can be used to derive climatological averages of aerosol direct radiative forcing. The work of Putaud et al. (2004 and 2010) and Van Dingenen et al. (2004) gathered information from long and medium term observations from rural, near-city, urban, and kerbside sites in Europe to highlight similarities and differences in aerosol

characteristics across the European network. As more sites provided access to longer data sets, the next series of papers (2010 up to present) addressed the issues of regional variability and trends with more robust statistical approaches and providing a comprehensive view of the aerosol variability to be used for model constraints.

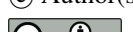



Variability for the in-situ climate-relevant aerosol properties relevant to SARGAN are documented for many GAW stations.

Integration of results from different sets of stations addressed different scales, from country (Sun et al., 2019) to continental (Sherman et al., 2015, Asmi et al., 2013; Fountoukis et al. 2014; Zanatta et al., 2016; Cavalli et al., 2016; Crippa et al. 2014; Pandolfi et al., 2018) to global (Collaud Coen et al., 2013; Asmi et al., 2013; Andrews et al., 2011; Andrews et al., 2019; Sellegri et al., 2019).

Generally, the seasonal variability of number concentration, and of the scattering and absorption coefficients, is much larger than diurnal variability at all sites (Sherman et al., 2015; Asmi et al., 2011) except at mountain observatories where meteorology plays a key role (Andrews et al, 2011; Collaud Coen et al., 2018). Typically, changes in aerosol intensive properties can be related to known sources. Timing of their maximum impact leads to well-defined seasonality that varies widely from site to site with the peak occurring at different times of year worldwide (e.g., Schmeisser et al., 2018). In

Europe, some aerosol properties at non-urban/peri-urban sites can be divided into different typologies connected to large geographical areas (i.e. Central Europe, Nordic, Mountain, Southern and Western European), for the different properties: carbonaceous aerosol concentration (Cavalli et al., 2016; Zanatta et al., 2016; Crippa et al. 2014); optical properties (Pandolfi et al., 2018); number concentration (Asmi et al., 2011); number of cloud condensation nuclei (Schmale et al., 2017) or chemical composition (Zhang et al., 2007; Crippa et al. 2014). This feature was used by Beddows et al. (2014), to propose a

representation of aerosol number size distribution in Europe with a total of nine different clusters for the whole continent. Two recent studies addressed variability for specific areas, using measurements from Arctic stations (Dall'Osto et al., 2019) and mountain stations (Sellegri et al., 2019). Interestingly, none of the studies detected statistically significant regional work-week or weekday related variation for any of the aerosol variables, indicating that the stations are relatively free from local emissions and that regional effects dominate over local effects.


Time series longer than a decade are generally required to derive trends and a lesser number of studies are available, in particular those integrating information from large sets of stations. Statistically significant trends in $\sigma_{sp}$ (decreasing), were found at 2 sites of NFAN in the US (analyzing trends from mid 90's to 2013) (Sherman et al., 2015). Similar results for a more globally representative set of sites were obtained for a comparison period of up to 18 years 1992–2010 (although less

for some sites) by Collaud Coen et al. (2013); for mostly European sites by Pandolfi et al., (2018) for aerosol optical properties (comparison period ending in 2015) and Asmi et al. (2013) for aerosol number concentration. Whenever a trend was detected, it was generally decreasing for the majority of the sites for almost all aerosol extensive variables. Exceptions (increasing trends) were found at several sites that could be explained by local features or by influence of emissions from the Asian continent. Decreasing trends have been reported in the literature for columnar AOD as well (e.g., Yoon et al., 2016;

Zhao et al., 2017; Ningombam et al., 2018; Sogacheva et al., 2018). Decreasing trends in number concentration are explained by reduction of anthropogenic emissions of primary particles, $SO_2$ or some co-emitted species, as also shown by



Aas et al., (2019) for sulfur species and Tørseth et al (2012) for PM10, PM2.5 and sulphate. In particular, Tørseth et al. (2012) show strong decreases, ca 50%, in the period 2000 to 2009 in PM10 and PM2.5. Decreasing trends (of the order of a few %/year for all variables were more pronounced in North America than in Europe or at Antarctic sites, where the majority
of sites did not show any significant trend (e.g., Collaud Coen et al., 2013).

The difference in the timing of emission reduction policy for the Europe and North American continents is a likely explanation for the decreasing trends in aerosol optical parameters found for most American sites compared to the lack of trends observed in Europe. In fact, the decreasing trends in Europe for aerosol optical variables were more detectable in
Pandolfi et al. (2018) using a 2000-2015 analyzing period than in Collaud Coen et al. (2013) using a comparison period of a maximum of 18 years ending in 2010. These studies did not find a consistent agreement between the trends of N and particle optical properties in the few stations with long time series of all of these properties; this is partly explained by the fact that aerosol light scattering coefficient is dominated by a different part of the aerosol size distribution than number concentration, and hence the two parameters are likely to have different sources.


The analysis of trends in aerosol properties needs to be regularly revisited as longer homogeneous time series become available at more sites, providing better spatial and temporal coverage. As shown in previous studies, trend and variability studies of aerosol properties still face some limitations due to heterogeneous time series, local effects that can only be addressed by some degree of redundancy among GAW stations, etc. It is also important to note that trends in terms of both
statistical significance and sign are very sensitive to the period and the methodology used for the calculation. The fact that different aerosol variables show opposite trends at some sites also suggests that further analysis is needed to better understand how the different aerosol parameters are connected to each other in the long term. These studies highlight the fact that other than in Europe and North America, and a few Antarctic stations, no trends can be derived due to lack of data from many areas in the world, as mentioned by Laj et al. (2010) 10 years ago!


Several studies have recently used in-situ measurements from, among others, the GAW network for a broad evaluation of the models, in particular in the framework of the AeroCom initiative (https://aerocom.met.no/):

- Particulate organic matter concentration: Tsigaridis et al. (2014) have found for 31 AeroCom models, compared to remote surface in-situ measurements in 2008-2010, a median normalized mean bias (NMB) underestimate of 15%
for particulate organic carbon mass and an overestimate of 51% for organic aerosol mass. This would indicate OA/OC ratio in the models is too high, however, it is generally rather low and close to 1.4. While the bias values are robust at the sites investigated, it is assumed that the measurement data available at the time were not representative enough to provide robust global bias estimates for the models in question.

- Dust concentration: Huneeus et al. (2011) have used a set of dust measurements from the SEAREX/AEROCE
networks which are very valuable due to their global extent and harmonised data. 15 AeroCom models generally




overestimate the remote site surface concentrations within a factor of 10. However, they underestimate the magnitude of major dust events e.g., in the Pacific. Kok et al. (2017) suggest from comparison to in-situ measurements of dust size distribution, among other parameters, that AeroCom models do not have a sufficient coarse dust component, which suggests that dust may even have a warming direct radiative effect.

- Sulphate concentrations: The downward trends 1990-2015 of observed and modelled surface sulphate surface concentrations in the Northern hemisphere have been shown to be very consistent by Aas et al. (2019), using 6 AeroCom models and a unique large collection of network data across Europe, North America and Asia. The work convincingly shows the mitigation success of $SO_2$ emissions, which is only possible because of harmonised in-situ measurements.

- Particle number and particle size distributions: 12 AeroCom models with aerosol microphysics simulation capability were evaluated by Mann et al. (2014) in terms of total particle concentrations and number size distributions. Particle number concentrations were collected from 13 global GAW sites operating for 5-25 years, while size distributions were mainly from European sites of ACTRIS in the years 2008/2009. Number concentration was underestimated by the models by 21% on average.

- CCN concentrations: Of even more relevance for aerosol cloud radiative effects is the evaluation cloud condensation nuclei. 16 AeroCom models were evaluated by Fanourgakis et al. (2019) against measurements of CCN at 9 surface sites in Europe and Japan. A model underestimation of about 30% was found, depending on dry size and supersaturation assumed and season (larger underestimate in winter).

**5 Current status of the SARGAN station network**

**5.1 An overview of networks and organisations contributing to SARGAN**

As mentioned previously, the data provision is organized independently resulting in a rather complex system where data originates from WMO/GAW Global, Regional, and contributing partner stations which themselves belong to one or more networks, depending on the station history and funding schemes. For example, many stations are labelled simultaneously as GAW, ACTRIS and EMEP in Europe, or GAW and NOAA in the US. Information on station status can be found in the
GAW information system (GAWSIS). Registration to GAW does not exclude participation in other networks, either contributing to GAW or not. WMO/GAW report #207 (2012), reviewed the situation with respect to the different aerosol networks operating globally. Although data for the report were collected in 2009-2010, the current situation is quite similar to 10 years ago.

According to the GAW information system (GAWSIS, http://www.wmo.int/gaw/gawsis/), as of June 2019 the GAW aerosol network consists of 33 'Global' Stations', which are encouraged to participate in all the GAW measurement programmes and approximately 250 regional or contributing stations. Not all GAW stations are able to measure all aerosol variables listed in

Table 1 and SARGAN is, therefore, a subset of stations in GAW. Contributors to SARGAN consist primarily of these international networks and research infrastructures:

- NOAA-FAN (Federated Aerosol Network, https://www.esrl.noaa.gov/; Andrews et al., 2019) that consists of 7 stations located in the US and in 22 additional locations Worldwide in 2017. NOAA-FAN documents three SARGAN variables: $\sigma_{sp}$, $\sigma_{ap}$ and CN. EBAS hosts data from all NOAA-FAN sites (except WLG); aerosol data from NOAA baseline stations are also available from NOAA's ftp site.

- ACTRIS (Aerosol Clouds and Trace Gases Research Infrastructure, https://www.actris.eu/) that consist of 36
  stations, of which 5 are located outside Europe. ACTRIS documents all four SARGAN variables: $\sigma_{sp}$, $\sigma_{ap}$, CN, and PNSD that are accessible at http://ebas.nilu.no. The European Monitoring and Evaluation Programme (EMEP) recommends the measurement of most SARGAN variables in its monitoring strategy and some ACTRIS in situ stations are collocated with EMEP sites. For the four SARGAN variables the quality control procedures are operated in the context of ACTRIS. These data sets are often jointly labelled ACTRIS/EMEP, and all ACTRIS and
  EMEP data are accessible through the EBAS data portal, undergoing same data curation and quality control at the data centre.

- In addition to the two main contributors, other operating networks have provided information for the paper. These are the Interagency Monitoring of Protected Visual Environments (IMPROVE) in the US (http://views.cira.colostate.edu/fed/QueryWizard/Default.aspx), the Canadian Air and Precipitation Monitoring
  Network (CAPMoN) in Canada, the Acid Deposition Monitoring Network in East Asia EANET (http://www.eanet.asia/) in East Asia, the Korea Air Quality Network (KRAQNb) in South Korea and various individuals and data from smaller national or regional networks including the German Ultrafine Aerosol Network (GUAN) in Germany (http://wiki.tropos.de/index.php/GUAN).

Historically, there has been limited interaction among the different networks Worldwide, as mentioned in the WMO/GAW report #207 (2013). However, on the specific issues of monitoring short-lived climate forcers, the main contributing networks to GAW have managed to integrate many pieces of the data value-chain, from SOPs, to QA/QC and data access. Data sets have also been jointly exploited in several papers (Asmi et al., 2013; Collaud Coen et al., 2013; Andrews et al., 2011; Pandolfi et al., 2018; Zanatta et al., 2016; Andrews et al., 2019; etc...).

**5.2 An overview of networks and organisations contributing to SARGAN**

All sites are established with the intention of operating in the long term. For registration to GAW (Global or Regional status) a period of successful performance of typically three years is required before a new site is added. All sites are long term in nature and, for most, adhere to rigorous siting criteria that aim to avoid local sources as much as possible. Sites have been and continue to be selected to answer pressing scientific questions, which evolve with time, and to detect and attribute
changes in climate and climate forcing.



Currently, 89 different sites worldwide are contributing to the provision of at least one SARGAN variable. These sites are indicated in Figure 1 and Table 3. Note that they are potential additional collocated sites not used in this study. All information used to compile information for this study is directly derived from NOAA-FAN and ACTRIS/EMEP with

additional contributions from providers listed in Table 2. Except for a few sites, measurements from all sites comply with the quality assurance and data reporting criteria defined in Section 3.1 and 3.2. If the sites are part of a contributing network, inclusion is straightforward in that the contributing network will already have met the GAW quality control and data reporting criteria. We have allowed a few exceptions for some sites located in WMO regions I, II, III and IV to ensure the widest geographical coverage as possible.


Because of the specific purposes for which NOAA-FAN and ACTRIS/EMEP were established, the nature of the sites is clearly biased to provide information relevant on the regional scale. This is why urban and peri-urban sites are under-represented in SARGAN and that a majority of sites are sampling in environments far from local emission sources, with a station footprint that is generally quite large. The issue of spatial representativeness of observing stations has been addressed

in many papers (e.g., Wang et al., 2018; Sun et al., 2019), and in particular related to air quality monitoring (e.g., Joly and Peuch, 2012). Representativeness of a site describes how the measurements can be used to derive information for a given time or spatial scale, or for a given kind of environment. This information is key whenever ground-based observations are used to compare with space-based measurements or for evaluating models. However, defining station representativeness is not unambiguous and several papers exist with different definitions (Joly and Peuch, 2012).


Station representativeness is very often addressed using density plots identifying the most probable origin of air mass trajectories terminating at the station over a certain time (typically 3 to 6 days). Many stations in SARGAN can provide such analyses often performed to discriminate source areas influencing the site for climatological studies. Schutgens et al. (2017) discussed representativeness of ground-based observations both in terms of spatial and temporal averaging showing that

significant errors may remain even after substantial averaging of data. Joly and Peuch (2012) developed a methodology to build a classification of European air quality monitoring sites, mostly based on regulated pollutants.

In this paper, site characterization is made with a two-criteria approach: 1) a criterion describing the main geographical setting (e.g., polar, continental, coastal, mountain) and 2) a criterion providing indications about the dominant footprint (e.g.,

forest, rural, desert, urban, pristine, regional background, mixed). Additional details on some of these categories are warranted. Mountain sites are not classified solely based on elevation (for example, high plateaux such as SPO and SUM are not considered mountain sites) but rather on the fact that the station is located higher than the surrounding environment.



For the air mass footprint, "Mixed" is used whenever no dominant air-mass footprint criterion is identified. This is often the
655 case, for example, for mountain sites where air sampled during night differs from air sampled during day, due to local
orographic effects. "Pristine" is used whenever the site is located far away from any anthropogenic or natural sources.
Obviously, no simple site characterization can completely capture the influences on a location and we are aware of the
shortcomings of this classification. In the context of the paper, this simplistic scheme was considered the easiest way to
organize the statistical results. It should be mentioned that site characterization relies on authors' knowledge of the sites,
660 along with indications by the corresponding PIs.

### 5.2 Evolution of data provision in SARGAN

In their 2013 papers, Collaud Coen et al. (2013) and Asmi et al. (2013), evaluated trends in aerosol optical and physical
properties based on times series extending from 1993 to 2010. At that time, 24 sites worldwide had the capacity to provide a
≥10 year time series for at least one of the optical or physical properties. In 2018, there are 52 stations capable of providing
665 ≥10 year time series for optical or physical properties. The increase in number is clearly driven by many European sites
initiated between 2000-2005, in particular through ACTRIS, but there are also now multiple sites in Asia with 10 year time
series through collaboration with NFAN. Figure 2a, b, c and d illustrate the evolution of data provision in SARGAN for
optical, and physical properties.

670 Globally, considering all four variables, there has been a very significant improvement of data provision in the last 10 years,
with almost five times more stations operational as shown in Figure 3. In 2017, the status is that for absorption there are 50
sites with 1 year of data, 37 sites with 5 years of data, and 20 sites with 10 years of data. For scattering, the parallel
development is: 56 sites with 1 year, 45 sites with 5 years and 30 sites with 10 years of data.

675 It is worth noting that, besides Antarctic stations, no stations were located outside North America and continental Europe in
Collaud Coen et al. (2013) and Asmi et al. (2013), while 9 stations outside those regions are now contributing to Collaud
Coen et al. (submitted). Overall, the total number of measurement-years increased substantially which will contribute to a
more robust vision of the state of the atmosphere. It remains a fact, however, that the number of stations providing
information in many areas (Africa, South America, Australia) is too low to draw overarching conclusions about trends for
680 those regions.

The number of stations would have been even higher except that a few were either closed between 2012 and now or moved.
This is the case for Mukteshwar station with the longest time series in India (2007-2015) which was moved in order to obtain
measurements at another location, thus interrupting the time series. This is also the case for Vavihill station (VAV) in
685 Sweden, moved to another location (Hyltemossa) in order to colocate aerosol and greenhouse gases observations, and
Southern Great Plains (SGP) which shifted buildings and instruments and left the NFAN in September 2017. Other stations



actually closed (e.g., THD (June 2017) and SMO (July 2017)). CPR was offline for many months due to a hurricane (September 2017-March 2018), and GSN has only very sparse data (not usable for trend analysis) since 2016 due to monsoon damage. The Global GAW station of NCO-P in Nepal also stopped operating in 2016. Closure of some important
stations in regions where measurements are lacking is clearly unfortunate in the context of SARGAN.

The access to data through the GAW-WDCA database EBAS has been monitored since May 2009. The use is extensive, both in volume, number of users and geographical distribution of download of data. The users of GAW-WDCA data are distributed worldwide. In the period between May 2009 – October 2019, 4110 unique client IPs from 72 different countries
have downloaded data, each of them accessing the databases from one to numerous times. Note that some large research institutes (e.g., NOAA in US) have 1 single IP for all users. In total, more than 125,000 full measurement years of data have been downloaded from GAW-WDCA since May 2009. The development over time is shown in Figure 4 with a strong increase over time.

## 6 Present-day variability of aerosol physical and optical properties derived from SARGAN stations

**6.1 General criteria for data selection**

The present article provides an updated overview of the distribution of aerosol properties based on the information available in EBAS from sites listed in Table 3. The analysis is based on data collected in 2017 to provide the most updated view of measurements worldwide. The analysis is restricted to a very basic statistical overview (yearly and seasonal median, percentiles, average) that is completed, for some stations, by the trend analysis performed as part of Collaud Coen et al.
(submitted). To perform this analysis, we preferentially used data collected in 2017. In case the coverage for 2017 was insufficient (see criteria below), data from 2016 was used. This is indicated in tables SM1 and SM2.

All sites contributing to SARGAN in 2017 were included in the analysis. The analysis is based on hourly data of $\sigma_{sp}$, $\sigma_{ap}$ and PNSD. Only validated measurements were used, i.e. data following the curation described in section 3.2, and, for an aerosol
parameter, the datasets from the different stations were further harmonized (e.g. to ensure that the time-vectors and data were of the right format and comparable with each other). Prior to the calculation of the summary statistics, a few problematic data points were also removed, following communication with the PI. For each site, annual and seasonal summary statistics were computed (median, 10th and 90th quantiles); the results were included only if 75% of the hourly data was available over the statistics reference period (with the exception of BRW, MLO and SPO whose respective coverage for each aerosol
property is detailed in tables SM1 and SM2). In cases where the 2017 coverage was not sufficient (i.e. <75% for all seasons) for an aerosol parameter (e.g., due to instrument failure or natural disaster impacting the station), the 2016 data was considered for that parameter. In cases where the coverage for that aerosol property was insufficient also for 2016 (i.e. <75% for all seasons), the site was discarded from the analysis for that aerosol property. For the sake of simplicity, the seasons





were attributed using the common division December – February, March – May, June – August and September – November

at all sites, even for the stations where other temporal divisions would be more relevant. This is, for instance, the case for CHC, where meteorological conditions are affected by two main seasons (May – September and December – March) with tropical characteristics (i.e. dry and wet, respectively). For all station types and time scales (year and seasons), the discussions are limited to the sites where data availability was sufficient, and which statistics are shown in the relevant figures and tables.


As mentioned in Table 3, many sites are actually influenced by different air-mass types, and some of them are influenced by anthropogenic sources. For most sites, data from all air masses are included in the statistical analysis. For BRW, MLO and SPO, the data included in this overview do not include all valid measurements collected at these three sites, but only the data corresponding to clean air masses. Clearly, in that case, the coverage criteria indicated above do not apply.  This screening

protocol, performed by the institutes operating the instruments, results in a lower annual data coverage and in a bias towards lower levels but ensures data consistency with the multi-decadal data available from these sites.

### 6.1 General criteria for data selection

### 6.1.1 Data Handling

Sixty-four sites in total contributed in 2016/17 to the SARGAN initiative by providing optical aerosol properties: 53 for

absorption and 55 for scattering coefficient data, respectively; for 29 of these sites was possible to compute also single scattering albedo. Four different types of filter-based absorption photometers were included in the analysis of $\sigma_{ap}$: the Multi Angle Absorption Photometer model 5012 (MAAP, by THERMO-Scientific Inc, USA), the Continuous Light Absorption Photometer 3-wavelengths (CLAP-3W, NOAA), the Aethalometer AE31 (Magee Scientific, USA) and the Particle/Soot Absorption Photometer 3-wavelengths (PSAP-3W, Radiance Research Inc). It is important to note that data from

Aethalometer AE33 (Magee Scientific, USA) were not used in this study as a unique value for converting the measured attenuation coefficient to particle light absorption coefficient ($\sigma_{ap}$) has not been fixed. The MAAP provides absorption at 637 nm (Mueller et al, 2011), the CLAP at 461, 522 and 653 nm (Ogren et al., 2017), the AE31 at 370, 470, 520, 590, 660, 880, 950 nm (Hansen et al., 1984) and the PSAP at 467, 530, 660 nm. Summary statistics for absorption were based on $\sigma_{ap}$ at 637 nm for MAAP and on $\sigma_{ap}$ at the wavelength closest to 637 nm for other instruments. At PDM the absorption was measured

by a single wavelength AE16 at 880 nm: at this site the statistics were based on absorption adjusted to 637 nm assuming a constant AAE = 1.

For aerosol scattering, the instrument deployed is primarily the Integrating Nephelometer 3563 (TSI Inc, USA), the Aurora 3000 (Ecotech Inc, AU) and the NGN-2 (Optec Inc, USA). The only exceptions are at PDM and SRT, where Aurora M9003

(Ecotech Inc, AU) nephelometers are utilized. Summary statistics for aerosol scattering coefficient were computed at the





wavelength closest to 550 nm for each instrument type, i.e. at 550 nm for the TSI and Optec nephelometers and at 525 nm for the Aurora 3000 and Aurora M9003. Due to the large dependence of scattering on hygroscopicity of aerosol, only scattering coefficients associated with a sample relative humidity less than or equal to 50% were used; this threshold, slightly higher than the prescribed 40%, allowed for more sites to be included, and was consistent with Pandolfi et al. (2018).


Single scattering albedo was computed at 550 nm using the optical properties closest to 550 nm for all multiple wavelength instruments. For $\sigma_{ap}$ by MAAP the data was adjusted to 550 nm assuming a constant AAE = 1.

For both $\sigma_{ap}$ and $\sigma_{sp}$, the effect of the difference in the instrument wavelength on the comparability of the data used for the
summary statistics was considered negligible; the only exception was for the estimate of $\sigma_{ap}$ at 637 nm by AE16 and of $\sigma_{ap}$ at 550 nm by MAAP, for which a constant AAE = 1 was assumed.

### 6.1.2 Global variability of optical properties

The variability of aerosol absorption and scattering coefficient medians is presented in Figures 5a and 5b and in Tables SM1 and SM2 along with other main summary statistics. The range of variability of both $\sigma_{ap}$ and $\sigma_{sp}$ is high, spanning several
orders of magnitude, with variability at least partly explained by a few main drivers: site latitude, site geographic location/footprint and the distance from the main anthropogenic sources. Globally the spatial variability of scattering and absorption has large similarities, being both featured by largest variability at mountain sites and minimum variability at urban polluted sites (e.g. LEI, IPR). Within the mid latitudes, absorption and scattering tend to increase from sites with a rural or forest footprint towards those in mixed and urban conditions. Polar sites, both in the Arctic and Antarctic, exhibits
the lowest $\sigma_{ap}$ and $\sigma_{sp}$, occasionally below instrumental level of detection (LOD) for absorption. Besides polar sites, lowest $\sigma_{ap}$ and $\sigma_{sp}$ values are generally observed at mountain sites, e.g. JFJ, ZSF and MLO (whose data is screened for clean air sector and may partly explain the low value), along with the Pacific coastal background site of CGO. A similar situation is observed for the lowest $\sigma_{sp}$ which, besides for pristine sites, are observed for mountain sites. Interestingly, the mountain site of JFJ, in Switzerland has a median $\sigma_{ap}$ and $\sigma_{sp}$ lower than a few polar sites, i.e. ALT, BRW, PAL, ZEP, and ALT, BRW,
NMY respectively.

The variability is generally higher at sites with low $\sigma_{ap}$ and $\sigma_{sp}$, reflecting the contrasting transport, in the case of pristine sites between the very low background values and the increase to advection of less clean air masses, and for mountain sites, the contrasting diurnal or seasonal transport patterns. A very good example is TIK, showing the largest medians among polar
sites, where $\sigma_{ap}$ spans over one order of magnitude, reflecting the collection of both clean and polluted air masses, most likely affected by biomass burning in the high latitudes.


The highest values and the smallest variability in both $\sigma_{ap}$ and $\sigma_{sp}$ are observed for urban/peri urban sites (e.g. LEI, UGR, IPR). It is interesting to note that occasionally the rural stations as AMY (East Asia) and KOS (Central Europe) have median

and range values of $\sigma_{ap}$ similar to urban sites, despite being located in rural areas far from local sources. PDI and BKT, both mountain sites in Southeast Asian tropical forests, exhibit large medians for both $\sigma_{ap}$ and $\sigma_{sp}$ compared to other forest/mountain sites due to recurrent impact by biomass burning (Bukowiecki et al., 2019). Similarly, biomass burning events related to anthropogenic emission from mainland China also affect via regional transport both LLN, another mountain site in SE Asia, and AMY.


At mountain sites in Southern Europe (MSA, HAC and CMN), a large scattering and absorption range is observed, comparable to that at rural background sites. This variability is partly due to the mixed nature of the sites, to long-range transport events (e.g., Saharan dust outbreaks, coal burning from Eastern Europe) and biomass burning both from forest fires in summer and domestic heating in winter. Saharan dust transport events partly explain the variability observed in other

Southern European sites, e.g. FKL.

The seasonality of $\sigma_{ap}$ and $\sigma_{sp}$ is presented in Figures 6a and 6b. The variability of the season median is much lower than the yearly variability reflecting the importance of transport in the variability. The most pronounced annual seasonality is observed at high mountain sites due to the seasonal variation of the boundary layer height and the local circulation induced

by thermal winds that follow the ground temperature cycle. In the case of mountain sites, the seasonality is also reflecting the index of boundary layer influence as defined by Collaud Coen et al. (2018). Generally, seasonality is largest at sites in an urban setting (e.g. UGR, NOA, LEI-M) and at those recurrently influenced by transport of either local or distant anthropogenic emissions (e.g. IPR, GSN). Also biomass burning can have large influence on absorption seasonality and on absolute levels, e.g., the Asian sites of GSN, LLN and AMY. In general, the seasonal variations are very clearly observed at

remote sites, for example at ALT and TIK, where the seasonality of air mass origin bringing high levels of aerosol during some parts of the year dominate the very minimal local emissions.

### 6.2.3 Global variability of single scattering albedo

For stations providing simultaneous measurements of scattering and absorption coefficients, it is possible to derive the single

scattering albedo which is done at 550 nm. Overall, $\omega_0$ is computed for 31 stations and presented in Figure 7. Median $\omega_0$ values range from slightly less than 0.8 to almost purely scattering particles with $\omega_0$ close to 1. The highest values are found at coastal and polar sites clearly influenced by inorganic salts and sulfur-rich particles. The lowest $\omega_0$ are observed at sites in southern Europe (IPR and UGR), which are impacted by desert dust, biomass burning and local emissions. Only 6 sites have median $\omega_0$ below 0.9 but only the coastal, mountain and polar sites exhibit 25th percentiles constantly above 0.9. Variability

of $\omega_0$ is strongly connected to air mass characteristics with, for a single station, a typical range of variability (25th-75th





percentile) of approx. 0.05 units of $\omega_0$. The variability at sites characterized as "Mixed", and in particular the mountain sites, is not higher than at other sites. The switch from free tropospheric air to boundary layer for the mountain sites does not appear to significantly affect $\omega_0$.

**6.2.4 Comparison with AeroCom model outputs for optical properties**

The AeroCom initiative has focussed since 2002 on the evaluation of global aerosol models with observations (aerocom.met.no). The integration of emission sources and aerosol processing leading to radiative effects requires complex models, which are increasingly coupled in high detail to general circulation models. Quantifying the climate forcing from aerosols requires a range of parameterised processes and derived properties of the global aerosol, which must be constrained

by observations. The atmospheric dispersion of the aerosol, their optical properties, the attribution to natural and anthropogenic sources, the potential of particles to influence clouds, and temporal trends – all these components need to be understood to quantify the radiative effect of aerosols. A network of in-situ aerosol measurements, well calibrated and available for long-term trend characterisation will provide important insights into the ability of models to realistically compute these radiative effects.


The recent generation of AeroCom models has been asked to provide additional diagnostics on dry scattering and absorption coefficients at ground level. These are currently being analysed by the two companion papers of Gliβ et al. (submitted) and Mortier et al. (submitted) using 14 model simulations of present day (2010 emissions and meteorology) to construct an ensemble mean AeroCom model and aerosol information extracted from SARGAN surface sites. For a detailed analysis of

comparison for variability and trends, readers can refer to the two companion papers. Here we simply provide an overview of the AeroCom model ensemble with observations for the specific SARGAN sites. Figures 8a and b compare AeroCom mean model against the 2017 data of measured dry scattering and absorption coefficients for selected sites, as used above for figure 5. Mountain sites at altitude above 1000 m asl are excluded, because of missing model diagnostics at mountain tops.

Overall, the performance of the model ensemble varies greatly as a function of station location, for both scattering and absorption coefficients. Figure 8 compares observations and model ensemble results for the grid point corresponding to the station location. It shows a normalised mean bias of, on average, -28% between scattering by AeroCom models and observations, pointing to regional deficiencies in aerosol models. The normalised mean bias for absorption is lower (-18%) but still showing an underestimate by the AeroCom models. Obviously, there is, for both scattering and absorption, a large

station-to-station variability in the bias, showing either good agreement, under- or over- prediction depending on the site. There is also a significant variability of the normalised mean bias between models and observations when calculated for each season. This is also the conclusion of Gliβ et al. (submitted) which quantified the biases to -44% and -32% for scattering and absorption, respectively and listed possible causes for the biases such as overestimate of scattering enhancement due to





hygroscopic growth and the differences in the treatment of absorption optical properties of black carbon, dust and organic

aerosol. At this stage, additional investigations are needed to identify what accounts for the observed differences between

model and observations.

### 6.2.5 Observed and modelled trends of aerosol optical properties

The issue of long-term trends for the aerosol in-situ optical properties is specifically addressed in Collaud Coen et al. (submitted) using data from WDCA extending back to 40 years for some stations. Collaud Coen derived time series of

measured scattering, backscattering and absorption coefficients as well as the derived single scattering albedo, backscattering fraction, scattering and absorption Angström exponents at stations with at least 10 years of continuous observations. With respect to the previous trend assessment (Collaud Coen et al., 2013) which used data extending up to 2010, the number of stations with time series longer than 10 years has almost doubled (24 in 2010, 52 currently) so that the spatial coverage is improved and various additional environments are covered in Europe, North America and in polar regions. The few stations

in Asia, Africa, South America and in Oceania/Pacific region cannot, however, be considered as representative for their continents/regions, both because of their small number and also because mountainous and coastal environments are overrepresented relative to the continental environment with rural, forest or desert footprints.

Methodologies and results are presented in detail in Collaud Coen et al. (submitted) and are simply summarized here for

scattering and absorption coefficients as well as single scattering albedo (Figure 9). For scattering coefficient, statistically significant (ss) increasing trends are found at polar and coastal stations with rural background, pristine and forest footprints, whereas the largest ss decreasing trends are primarily found at stations with mixed and urban footprints. Few mountainous stations have a ss scattering coefficient trends, whereas all of them have ss decreasing absorption coefficient trends. Almost all stations have either ss decreasing or not ss trends in the absorption coefficient; the stations with increasing trends are

influenced by polar or rural background footprints. The single scattering albedo trends seem not to be dependent on either the environment or on the footprints, but rather on the geographic area (Collaud Coen et al., submitted).

Analysis of the long-term information provides evidence that the aerosol load has significantly decreased over the last two decades in the regions represented by the 52 stations. Currently, scattering and backscattering coefficients trends are mainly

decreasing in Europe and North America and are not statistically significant in Asia. Polar stations exhibit a mix of increasing and decreasing trends. A few increasing trends are also found at some stations in North America and Australia. Absorption coefficients also exhibit mainly decreasing trends. Generally, these decreases in aerosol burden are expected to be a direct consequence of decreases in primary particles and particulate precursors such as $SO_2$ and $NO_x$ due to pollution abatement policies.




The single scattering albedo is one of the most important variables determining the direct radiative impact of aerosol so that its trend analysis - derived for the first time from a large number of stations - has the largest climatic relevance. The global picture is nuanced with ss positive trends mostly in Asia and Eastern Europe and ss negative trends in Western Europe and North America leading to global positive median trend of 0.02%/y. 15 stations exhibit a positive single scattering albedo

trend (relatively more scattering) while 9 stations exhibit a negative trend (relatively more absorption).

Trends in scattering and absorption coefficients are also estimated by Mortier et al. (submitted) using AeroCom and CMIP6 models that have simulated the historical evolution of aerosol properties. For both variables, simulated trends are in agreement with SARGAN derived trends suggesting significant decreases found over North America and Europe, although

the number of models providing trends in $\sigma_{ap}$ and $\sigma_{sp}$ remains limited. Comparison with observations is also restricted to sites below 1000 m asl which further reduces data points for comparisons. However, decreasing trends in AOD and sulphate are observed for North America and Europe for both model and observational data. Asian in situ surface data are too sparse to derive a regional trend for that region but it is worth indicating that not statistically significant AOD and sulphate trends are found in the overall period 2000-2014 over southern and eastern Asia. This suggests that there are different trends in aerosol

burden between North America and Europe and Asia. From model data alone, a global trend can be derived. Globally, the average model trend for 2000-2014 amounts to an increase of +0.2 %/yr for $\sigma_{sp}$ and +1.5%/yr for $\sigma_{ap}$, respectively, higher than what is observed at ground-based stations.

In addition to evaluating trends for the overall time series, Collaud Coen et al., (submitted) analyzed the evolution of the

trends in sequential 10y segments. For scattering and backscattering, statistically significant increasing 10-yr trends are primarily found for earlier periods (10-yr trends ending in 2010-2015) for polar stations and Mauna Loa. For most of the stations, the present-day statistically significant decreasing 10-yr trends of the single scattering albedo were preceded by not statistically significant and statistically significant increasing 10-yr trends. The effect of abatement policies in continental North America is very obvious in the 10-yr trends of the scattering coefficient that shift to statistically significant negative

trends in 2010-2011 for eastern and central US stations.

There are some discrepancies between the work of Collaud Coen et al. (submitted) and Mortier et al. (submitted) in particular regarding trends derived for specific regions. This may result from different methods used to aggregate measurements to long time series, or to differences in the time period (2000-2018 versus 2009-2018) but, overall, they both

confirm the shift of polluting activities from the developed countries to the developing countries during the last two decades and may also demonstrate the relatively higher reduction of BC-rich emission in some regions, which will affect aerosol forcing estimates.





### 6.3 Global distribution of aerosol physical properties

#### 6.3.1 Data Handling

Data collected at 57 sites contributing to SARGAN were analysed to provide an overview of the condensation nuclei in the atmosphere. Measurements are performed with condensation particle counters (CPC) and mobility particle size spectrometers (MPSS); note that when both CPC and MPSS were concurrently run at a site, only MPSS data were included in the analysis, as it allowed additional investigation of the PNSD. For MPSS measurements, data inversion was performed by the institutes operating the instruments, and, for both CPC and MPSS, particle number concentrations were reported in particles per cubic centimetre at STP, i.e., T = 273.15 K and P = 101 300 Pa, following the recommendations from Wiedensohler et al. (2012). As discussed in the overview of European PNSD and CN conducted by Asmi et al. (2011), the diameters associated with MPSS data correspond to the geometric mean diameter of the size intervals used in the inversion. MPSS measurements are moreover usually representative of dry aerosol properties, as the operating procedures described in Wiedensohler et al. (2012) indicate that the relative humidity of the sample air should be kept below 40%. In total, after excluding the datasets with insufficient data availability (with respect to the criteria reported in Section 5.1), CPC measurements collected at 21 stations and MPSS data from 36 sites were included in the analysis (Table SM3 in the Supplementary).

To allow for the comparison of CN values derived from both instrument types, particle concentration in the range between 10 and 500 nm was inferred from MPSS measurements and assimilated to total CN (hereafter referred to as $N_{tot}$). This size range was selected as it was common to most of the MPSS included in this study. In addition, the lower end of this size range is comparable to the lower cut-off diameter of 14 of the 21 CPCs involved in the comparison (10 or 11 nm), and we assumed that particles larger than 500 nm only contributed little to $N_{tot}$. The legitimacy of this approach was supported by the fair agreement between $N_{tot}$ derived from collocated CPC and MPSS measurements at several sites. Moreover, using available MPSS data, we found that, on average, particles in the range between 10 and 11 nm contributed less than 1% to $N_{tot}$ (90th percentile of the contribution: 5%), suggesting that such small cut point difference was not a major issue for $N_{tot}$. However, the influence of a larger difference in lower cut points could not be discounted; this was, for instance, the case for ETL, ARN and GSN, where particles down to 2.5 nm were accounted for in $N_{tot}$ (CN data were collected with a CPC TSI 3776 at these sites).

Results in the next section are discussed with respect to the classification of the stations reported in Table 2, including both the geographical and footprint criteria. Also, in order to describe the time evolution of CN and PNSD across the year, observations are categorized by seasons. Diurnal variations were not studied here, but would be expected to be strong for certain site types and conditions (e.g., mountain upslope/downslope, urban local traffic, etc.).



### 6.3.2 Global variability of physical properties at SARGAN sites

As shown in Figure 10 and Table SM3, the lowest particle concentrations are typically observed under conditions of minimal anthropogenic influence, at polar sites, where yearly medians of $N_{tot}$ are of the order of $10^2$ cm$^{-3}$. Overall, as discussed earlier
by Asmi et al. (2011), these stations also display a very clear seasonal cycle compared to other geographical categories, with a summer maximum of $N_{tot}$ likely resulting from both enhanced secondary aerosol formation, including new particle formation (NPF), and transport (Croft et al., 2016; Nieminen et al., 2018).

In contrast with polar sites, stations located in urban areas, both continental and coastal, exhibit the highest $N_{tot}$, with yearly
medians in the range $10^3$-$10^4$ cm$^{-3}$. These sites, all located in Europe, also display a less pronounced seasonal variation (Figure 11). Slightly greater median values are, nonetheless, observed during summer, when the atmospheric boundary layer (ABL) height is also increased relative to colder seasons. This suggests the presence of an additional source of aerosols in summer which compensates for the ABL height dilution effect, as recently discussed by Farah et al. (submitted) who moreover suggested a photochemical or biogenic source. The overall weak seasonality observed in lowland urban areas is
likely related to the contribution of very local sources which do not have any strong seasonal cycle (e.g., traffic). The local nature of the observations collected at urban sites is supported by the differences between the measurements performed at neighbouring sites (e.g., LEI and LEI-E).

Remaining sites, including mountain and non-urban continental and coastal stations, do not exhibit as clear a common
behaviour as the sites located at high latitudes or in urban areas. They display, on average, intermediate $N_{tot}$, with yearly medians of the order of $10^2$-$10^3$ cm$^{-3}$. The signature of their dominant footprint is clear, with lower concentrations and stronger seasonal contrast observed in forested areas compared to rural background stations, while the distinction between the different geographical categories is in contrast less evident. Nonetheless, in agreement with previous observations from Asmi et al. (2011), particle concentrations measured at mountain sites tend to be lower compared to nearby lowland sites
(e.g. SNB vs KOS). Mountain sites, and in specific those characterized by mixed footprints, tend to exhibit somewhat more pronounced seasonality relative to lowland stations. This likely results from the strong impact of ABL height variability which, together with the topography of the sites, governs the concentration of particles and their precursors transported at high altitudes (Collaud Coen et al., 2018). Specifically, the summer enhancement of $N_{tot}$ observed at most of the mountain sites is certainly tightly connected to the increased frequency of ABL injections during this time of the year (e.g., Herrmann
et al., 2015). Apart from the lower concentrations, observations collected at non-urban continental and coastal sites display similar seasonal variations as in urban areas, which are again likely explained by the concurrent variability of particle sources and ABL dynamics.





In short, particle concentrations are overall higher during warmer seasons at all sites as a result of enhanced sources, in
connection with ABL dynamics for mountain sites. In addition, based on available MPSS data, the major contribution of
Aitken mode particles (30-100 nm) to the total particle number concentration also appears as a common feature of all
environments. In contrast, the magnitude of the seasonal cycle of $N_{tot}$, together with the variations of the PNSD, exhibits
some distinctive behaviour for the different geographical categories and footprint classes, with additional site-dependent
characteristics. However, among other factors (including the nature and proximity of the particles sources), the level of
anthropogenic influence appears to strongly affect the observations.

## 7 Using SARGAN for global climate monitoring applications

Climate observations are fundamental to many aspects related to prediction of future environmental changes and to meet the
requirements of the UNFCCC and other conventions and agreements. The establishment of a global network of observations
for assessment of atmospheric composition changes, adaptation to climate change, monitoring the effectiveness of policies
for limiting emission of pollutants and/or developing climate information services must define the specific observational
requirements for efficiently addressing these issues.

### 7.1 Response of SARGAN to GCOS principles

Measurement harmonization procedures allowing for direct comparison of data provided, together with the quality control
and quality analyses performed all through the data provision chain have considerably improved the value of SARGAN as an
essential piece of the in-situ segment of Earth Observations for its specific climate-relevant variables. SARGAN addresses to
all 10 basic principles of the WMO-IOC-UNEP-ICSU Global Climate Observing System (GCOS). GCOS is designed to
meet the requirements for climate observations which are essential to climate monitoring and support implementation of
UNFCCC and other climate conventions and agreements.

Considering the importance of aerosol properties in the Earth Climate system, it is important to define the GCOS
requirements for a number of variables that are, or may be in the future, defined as essential climate variables. Today, there
are four aerosol GCOS ECV products: AOD, Single-Scattering Albedo, Aerosol Extinction Coefficient Profile and Aerosol
Layer Height. Only Single-Scattering albedo is directly connected to SARGAN although the GCOS aerosol variables are
currently being revised to include ECVs connected to aerosol size, composition and hygroscopic properties. In its current
state SARGAN is able to address the ten basic GCOS Climate Monitoring Principles as follows (Table 4):

These requirements must include the spatial and temporal resolution of the observations, and their accuracy, precision, and
long-term stability. For each requirement, one additional specification is required to identify 1) Threshold or minimum





requirement defined as the value that has to be met to ensure that data are useful, 2) Goal or maximum requirement defined as the value above which further improvement gives no significant improvement in performance or cost of improvement would not be matched by a corresponding benefit likely to evolve as applications progress. In between « Threshold » and « Goal », « Breakthrough » is defined as an intermediate level that would lead, if implemented to a significant improvement for the specific application.


It is clear that requirements are defined for specific application areas, in this case climate monitoring applications as defined in OSCAR (https://www.wmo-sat.info/oscar/applicationareas). The Climate Monitoring application area is defined as such: "The WMO-IOC-UNEP-ICSU Global Climate Observing System (GCOS) is an internationally coordinated network of global observing systems for climate, designed to meet the requirements for climate observations, which are essential to

climate monitoring. Climate observations are fundamental to detect, model and assess climate change, support adaptation to climate change, monitor the effectiveness of policies for mitigating climate change, develop climate information services, promote sustainable national economic development and meet other requirements of the UNFCCC and other conventions and agreements". Observational requirements for other application areas have been recently published (Benedetti et al., 2018) or are currently underway as part of the WMO/GAW activities.

**7.1 Response of SARGAN to GCOS principles**

With the specific definition, and considering the results presented in this paper, in companion SARGAN papers and in previous studies, the following requirements can be defined for SARGAN variables.

The threshold for spatial requirements in the horizontal scale for SARGAN can be defined as the distance between two

observing points above which no redundancy is observed when measurements are performed in parallel. A few papers have addressed this issue by investigating the autocorrelation function between time series for different aerosol properties (Anderson, 2003; Sun et al., 2019) and they both lead to similar results related to observations at the ground: temporal variations of an intrinsic aerosol variable observed at the ground are no longer statistically correlated when stations are located more than several hundred km apart. To be more specific, Sun et al. (2019) suggest that correlation of absorption

coefficient time series from stations located 500 km apart is still approximately 0.5. A similar result is found for particle number in the 200-800 nm range, while distance for a similar correlation of 0.5 for particles in the lower size range (10-30 nm) is of the order of 100 km. This, of course, depends on several parameters including the intensity of emissions surrounding the station, and efficiency of removal rates (dry and wet deposition). Interestingly, similar temporal correlations are observed in IAGOS (In-flight Atmospheric Global Observing System) for aerosol variables in the upper atmosphere

(Ulrich Bundke, personal communication).





It is fair to consider that two stations located more than 1000 km apart will, therefore, for aerosol variables relevant to SARGAN, provide very little redundancy in their observations, especially if the stations are located over land. Assuming an advection velocity of 20 km h$^{-1}$, 1000 km would correspond to approximately 2 days, which is shorter than the aerosol

typical lifetime over continents. For observations over the oceans, it is clear that a larger threshold could be considered, corresponding to a turn-over time of approximately a week (i.e. several thousands of km). The threshold for the observation of climate-relevant parameters in SARGAN can, therefore, reasonably be set at 1000 km, while breakthrough and goals for the spatial resolution can, accordingly, be set at 500 km and 100 km, respectively. A 100 km spatial resolution would serve the purpose of deriving radiative forcing estimates at scales typical of a large urban area, together with providing information

extremely relevant for model and space-based observations. These indicated horizontal requirements for threshold, breakthrough and goal would require models to provide information on approx. 0.5°x0.5° degree resolution grids for goal, which is now often achieved.

Considering a total land-area in Europe of approx. 10 M km$^2$ (thus only including the Russian territory in geographical

Europe), and 63 measurement stations in operation (see Table), the measurement density in Europe is close to requirements for « breakthrough ». It is even close to the « goal » level if Russia is not considered. In North America, it is close to «threshold» (28 stations for 24 M km$^2$) and between recommended values for threshold and breakthrough for US territory only, including Alaska  (21 stations over approx. 10 M km$^2$). For all other regions of the World, the situation is below that recommended for minimal sampling, illustrating the huge gaps in network density.


Because SARGAN is based on individual observation points at the surface, the issue of vertical resolution is not relevant. However, the value of measuring both in the boundary layer and in the free troposphere is clear for many applications. Requirements for temporal resolution can be derived in a simpler way, considering that time-series datasets are often provided on a month-by-month variation in climate over long-time periods. Monthly data sets allow many variations in

climate to be studied and can be considered as threshold as long as the data is generated by representative original data sets. Information provided with a temporal resolution of one-day are suitable for addressing issues related to cloud cover, precipitation, impact of temperature, emissions, etc… and can be considered as breakthrough while the 1-hour resolution is a requirement for many applications such as estimating aerosol fluxes or radiative impact of aerosol plumes.

The maximum time lag between observations and the data being freely available is, for most applications, of the order of one year (threshold), although data providers are more and more requested to provide information on shorter timescale, with 24 hour delay and near-real-time (6 hour delay) corresponding to « breakthrough » and « goal » levels, respectively.

The definition of requirements for GCOS also asks to establish a level of uncertainty which accounts for all quantifiable

uncertainties. In the case of in-situ aerosol variables, requirements for the measurement uncertainties can be derived from the





observed variability on the different temporal scales, which is quite large. We have used suggested uncertainties provided in Table 2 for CN, $\sigma_{sp}$ and $\sigma_{ap}$. Uncertainties of $\omega_0$ is proposed following procedures of Sherman et al., (2015).

Stability is defined as the maximum permissible cumulative effect of systematic changes of the measurement system to allow long-term climate records compiled from assorted measurement systems. For the optical properties, Collaud Coen et al., (submitted) observed mainly decreasing trends for scattering and absorption coefficients in Europe and North America while no trend or a mix of increasing and decreasing trends are observed in other parts of the World. When statistically significant, trends derived by Collaud Coen et al. (submitted) for optical properties are of the order of a few (<2) %/yr maximum. This defines, for regions where trends are detectable, the threshold requirement for stability since expected trends

would not be detectable with higher stability values. Carslaw et al., (2010) have estimated the change in aerosol radiative forcing due to climate feedbacks in emission of aerosol precursors from natural systems. They show that a radiative perturbation approaching 1 $Wm^{-2}$ is possible by the end of the century. Detecting and attributing changes to a climate feedback due to changing natural emissions (wildfires, biogenic organic volatile compounds) would require a much lower uncertainty than currently achieved for CN, $\sigma_{sp}$ and $\sigma_{ap}$ and consequently $\omega_0$. At this stage, without more information on

trends, we are recommending values for stability of 1%/yr for breakthrough and 0.5%/yr for goal for all variables. Requirements for the GCOS application area for $\sigma_{sp}$, $\sigma_{ap}$, CN and $\omega_0$ are summarized in Table 5.

**8 Conclusions and future challenges**

The present article must be seen as the foundational framework for the observation of aerosol properties collected near-surface from ground-based stations Worldwide, in the context of GAW. SARGAN completes a ground-based aerosol

observing system composed additionally of the GAW associated networks GALLION and PFR. SARGAN relies on its regional constituents in the different WMO regions, of which ACTRIS in Europe and NOAA-FAN in the US are the principal contributors.

Although not fully implemented and operational, SARGAN sites share common methodological approaches for

measurement and data quality control, and a common objective to open access for all data, that are all defined as part of the Global Atmosphere Watch Scientific advisory group on aerosol. Data provision is currently operational with some sites providing information for more than several decades. The very strong motivation in the early 2000s to develop observations of aerosol climate-relevant parameters led to a substantial increase in operating ground-based stations and availability of data time-series with the required level of quality. We consider that the degree of integration of the different providers to

SARGAN has reached a mature level which has resulted in more and more users of the data worldwide.



The current SARGAN database can be used for many different applications. In this article, it is limited to very basic statistical descriptions, comparing variability of four SARGAN parameters at 89 sites and a preliminary approach to compare model and observations for the relevant variables. In the associated companion papers long-term climatological

trends are derived by Collaud Coen et al. (submitted) for the optical aerosol properties showing for the first time an unequivocal decrease of scattering and absorption coefficient in Europe, following a tendency already detectable in the US several years ago. Model studies (e.g., Mortier et al, submitted) find similar trends to the observations in North America and Europe. Open access to the SARGAN database should enhance the potential for many other applications. Analysis of trends for number concentration is already under way but we assume that SARGAN data can be efficiently used to support many

types of studies, related to aerosol impact on air quality, health or climate, quantification of emission sources or for the development of early-warning services.

The SARGAN initiative is currently limited to four variables that are directly observed. They are the only four climate-relevant aerosol variables measured near-surface for which a relatively consistent coverage exists worldwide. Providing

constraints on radiative forcing estimates would obviously require knowledge of trends and variability for other variables, such as aerosol chemical composition or number concentration of cloud condensation nuclei. Unfortunately, very few sites are currently including these variables in their observation program and they are mostly located in Europe as part of ACTRIS. It is clearly a huge and key challenge for the community to extend observations to additional variables, in particular for sites located outside Europe.


The distribution of sites providing information to SARGAN confirms the analysis made in many earlier reports and in Laj et al. (2010): a very strong bias still exists in the World data coverage, with Europe and the US well-represented and observations lacking in many other regions, in particular over WMO region III (Africa) and IV (Latin American and Caribbean), Russia, and large parts of Asia. Causes may be connected to difficulties making data accessible through the

World Data Centers in some cases, but for many areas of the World, it is directly related to lacking measurements. Detecting atmospheric trends of key atmospheric compounds requires long (>10 years) high quality records and, despite many initiatives, only a very few stations have managed to maintain operations for observing composition changes over more than a decade.

Laj et al. (2019) have recently proposed a series of recommendations to support atmospheric observations in emerging economies. Demonstrating how climate data/ information have direct relevance to policy making and explaining the local benefits that monitoring atmospheric composition changes bring to the country in terms of socio-economic impacts, in both the short and longer terms may help engage national stakeholders to commit to maintain and develop observation sites. Stimulating the demand for climate observations/ climate information of the kind provided by SARGAN at the user level in

the countries concerned would be absolutely important. The European concept of Atmospheric Research Infrastructures,



such as ACTRIS, was key to securing the necessary long-term engagement in the EU countries to support SARGAN observations. Similar approaches can be proposed, adapted to the different WMO regions.

In a recent comment in Nature, Kulmala (2018) suggested the establishment of 1,000 or more well equipped ground stations around the world tracking environments and key ecosystems, thus sampling beyond the observation of atmospheric composition only. Establishing observation sites with core measurement capabilities documenting key atmospheric components (greenhouse gases, reactive gases, aerosol properties) together with basic meteorology, operated by skilled personnel and providing access to measurement data in countries where this is still lacking would require a large scale coordinated effort that is far from being out of reach. Investments for atmospheric monitoring would be anywhere between

0,5 and 1 M US$ and annual operations between 50 and 100 kUS$ and 2-3 FTEP per site.

There is a growing number of multilateral climate finance initiatives designed to help developing countries address the challenges of climate change and air quality. They have a role in capacity building, research, piloting and demonstrating new approaches and technologies and are perfectly suited to be used for developing the needed atmospheric component of a

global Earth observing system. A "One Nation, One Station" approach to establish at least one reference stations in each country where information is lacking would definitely add essential information to large-scale modelling but also support local research, national policymakers, and promote business development for environmental services such as early warnings for extreme weather and atmospheric hazards.

**Acknowledgements**

Work performed within this paper received funding from ACTRIS-2 European Union's Horizon 2020 research and innovation programme under grant agreement No 654109.

NOAA base supports the following observatories: BRW, BND, MLO, SMO, SUM, SPO, and THD where efforts of the dedicated observatory staff and of programmer Derek Hageman are appreciates


IZO measurements are financed by AEMET. Effort and dedication of staff of the Izaña Observatory in maintaining the instruments is greatly appreciated. Acquisition and data curation was partially financed by European ERDF funds through different Spanish R&D projects of the Spanish Ministerio de Economía, Industria y Competitividad.

CPT acknowledges NOAA ESRL for their continued academic and technical support of the Cape Point Aerosol measurements, and the staff from the South African Weather Service who have contributed to the generation of the data records reported here.

Measurements at Welgegund are supported by North-West University, University of Helsinki and Finnish Meteorological
Institute.

PAL, SMR acknowledge support of, the Academy of Finland Centre of Excellence program (project number 272041), the
Academy of Finland project Greenhouse gas, aerosol and albedo variations in the changing Arctic (project number 269095)
and Novel Assessment of Black Carbon in the Eurasian Arctic: From Historical Concentrations and Sources to Future
Climate Impacts (NABCEA, project number 296302).

Aerosol measurements at Anmyon-do were supported by the Korea Meteorological Administration Research and
Development Program "Development of Monitoring and Analysis Techniques for Atmospheric Composition in Korea"
under Grant (1365003013). Measurements at Gosan were supported by the National Research Foundation of Korea
(2017R1D1A1B06032548) and the Korea Meteorological Administration Research and Development Program under Grant
KMI2018-01111.

The Lulin station is operated under the grants funded by the Taiwan Environmental Protection Administration.

Sites PDM, PUY, GIF, CHC and RUN are partially operated with the support of CNRS-INSU under the long term
observation program and the French Ministry for Research under ACTRIS-FR national research infrastructure. PDM and
GIF received specific support of French Ministry of Environment. ATMO Occitanie is acknowledged for sampling
operations at PDM. Measurements at SIRTA are hosted by CNRS and by the alternatives energies and atomic energy
commission (CEA) with additional contributions from the French ministry of Environment through its funding to the
reference laboratory for air quality monitoring (LCSQA). PUY acknowledges support from ATMO Auvergne Rhône Alpes
for sampling operations and the support from the personnel of the Observatoire de Physique du Globe de Clermont-Ferrand
(OPGC). The specific support of Institut de Recherche et Développement (IRD) in France and Universidad Mayor de San
Andres in Bolivia support operations at CHC operations.

Measurements at IMPROVE sites acknowledge support of US Environmental Protection Agency as the primary funding
source, with contracting and research support from the National Park Service. The Steamboat Ski Resort provided logistical
support and in-kind donations for SPL. The Desert Research Institute is a permittee of the Medicine-Bow Routt National
Forests and an equal opportunity service provider and employer. SPL appreciate the extensive assistance of the
NOAA/ESRL Federated Aerosol Network, of Ian McCubbin, site manager of SPLand of Ty Atkins, Joe Messina, Dan
Gilchrist, and Maria Garcia who provided technical assistance with the maintenance and data quality control for the aerosol
instruments.





Cape Grim Baseline Air Pollution Monitoring Station acknowledges the Australian Bureau of Meteorology for their long term and continued support of, and all the staff from the Bureau of Meteorology and CSIRO who have contributed to the generation of records reported here.


The aerosol measurements at the Jungfraujoch were conducted with financial support from MeteoSwiss (GAW-CH aerosol monitoring program) and from the European Union as well as the Swiss State Secretariat for Education, Research and Innovation (SERI) for the European Research Infrastructure for the observation of Aerosol, Clouds and Trace Gases (ACTRIS). The International Foundation High Altitude Research Station Jungfraujoch and Gornergrat (HFSJG) is acknowledged for providing the research platform at the Jungfraujoch.


The aerosol measurements at Kosetice has received funding from the European Union's Horizon 2020 research and innovation program under grant agreement No. 654109, and from the project for support of national research infrastructure ACTRIS – participation of the Czech Republic (ACTRIS-CZ - LM2015037) supported by the Ministry of Education, Youth and Sports of CR within the National Sustainability Program I (NPU I), grant number LO1415. The measurements were also supported by ERDF "ACTRIS-CZ RI" (No. CZ.02.1.01/0.0/0.0/16_013/0001315).


Measurements at the Madrid site have been funded by the following projects: CRISOL (CGL2017--85344-R MINECO/AEI/FEDER, UE), TIGAS-CM (Madrid Regional Government Y2018/EMT-5177), AIRTEC-CM (Madrid Regional Government P2018/EMT4329) and REDMAAS2020 (RED2018-102594-T CIENCIA). Measurements at Montsec and Montseny were supported by the Spanish Ministry of Economy, Industry and Competitiveness and FEDER funds under the project HOUSE (CGL2016-78594-R), by the Generalitat de Catalunya (AGAUR 2017 SGR41 and the DGQA). Aerosol measurements at El Arenosillo Observatory are supported by the National Institute for Aerospace Technology, by different R&D projects of the Ministerio Español de Economía, Industria y Competitividad (MINECO). Optical data acquisition would be not possible without the support of the NOAA/ESRL/GMD.



FKL and DEM acknowledge funding by the project "PANhellenic infrastructure for Atmospheric Composition and climate change" (MIS 5021516) which is implemented under the Action "Reinforcement of the Research and Innovation Infrastructure", funded by the Operational Programme ""Competitiveness, Entrepreneurship and Innovation"" (NSRF 2014-2020) and co-financed by Greece and the European Union (European Regional Development Fund).


At CMN, aerosol measurements have been partially supported by the Italian Ministry of Research and Education.

Measurements at Birkenes II are financed by the Norwegian Environment Agency.





Diabla Gora and Zielonka acknowledge the Data from Chief Inspectorate of Environmental Protection, State Environmental Monitoring. Measurements of absorption at IRB were performed by Aerosol d.o.o.; Ljubljana, Slovenia.

VAV acknowledges various Swedish FORMAS, Swedish Research Council (VR) grants and Magnus Bergvall and Märta och Erik Holmberg foundations and Swedish EPA for making the research possible at the VAV site.

NMY wishes to thank the many technicians and scientists of the Neumayer overwintering crews, which outstanding commitment enabled achieving continuous, high quality aerosol records over many years.
.

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



| Nomenclature | Definition |
|---|---|
| $\sigma_{ep}$, $\sigma_{sp}$[1], $\sigma_{ap}$[1] | The **volumetric cross-section for light extinction** is commonly called the particle light extinction coefficient ($\sigma_{ep}$), typically reported in units of $Mm^{-1}$ ($10^{-6}$ $m^{-1}$). It is the sum of the particle light scattering ($\sigma_{sp}$) and particle light absorption coefficients ($\sigma_{ap}$), $\sigma_{ep} = \sigma_{sp} + \sigma_{ap}$. All coefficients are spectrally dependent. |
| AOD[1,2] | **Aerosol optical depth**, defined as the integral over the vertical column of the aerosol particle light extinction coefficient. |
| $\omega_o$[2] | The **aerosol particle single-scattering albedo**, defined as $\sigma_{sp}/\sigma_{ep}$, describes the ratio of particle light scattering coefficient to the particle light extinction coefficient. Purely scattering aerosol particles (e.g., ammonium sulphate) have values of 1, while very strong absorbing aerosol particles (e.g., black carbon) may have values of around 0.3 at 550 nm. |
| AAOD | The **absorption Aerosol optical depth** is the fraction of AOD related to light absorption and is defined as AAOD=$(1-\omega_o)\times$AOD. |
| $g$, $\beta$ | The **asymmetry factor $g$** is the cosine-weighted average of the phase function, ranging from a value of -1 for entirely backscattered light to +1 for entirely forward-scattered light. The **upscatter fraction $\beta$** gives the fraction of sunlight scattered in the upwards direction (back to space), which depends on the solar zenith angle as well as the size distribution and chemical composition of the particles. |
| AE (or Å) | The extinction (scattering) **Angstrom exponent** is defined as the dependence of AOD (or ($\sigma_{sp}$)) on wavelength ($\lambda$), e.g., AOD$\propto C_0\lambda^{-AE}$ where $C_o$ denotes a wavelength-independent constant. The Angstrom exponent is a qualitative indicator of aerosol particle size distribution. Values around 1 or lower indicate a particle size distribution dominated by coarse mode aerosol such as typically associated with mineral dust and sea salt. Values of about 2 indicate particle size distributions dominated by the fine aerosol mode (usually associated with anthropogenic sources and biomass burning). |
| AAE | The absorption Ångström exponent (AAE) describes the wavelength variation in aerosol absorption. $\sigma_{ap}(\lambda)=C_o\lambda^{-AAE}$ where $C_o$ denotes a wavelength-independent constant. |
| MSCi, MACi | The **mass scattering cross-section (MSCi)** and **mass absorption cross-section (MACi)** for species $i$, often calculated as the slope of the linear multiple regression line relating $\sigma_{sp}$ and $\sigma_{ap}$, respectively, to the mass concentration of the chemical species $i$, is used in chemical transport models to evaluate the radiative effects of each chemical species prognosed by the model. This parameter has units of $m^2$ $g^{-1}$. |
| *f(RH), g(RH)* | f(RH) is the functional dependence of components of the aerosol particle light extinction coefficient ($\sigma_{ep}$, $\sigma_{sp}$, $\sigma_{ap}$) on relative humidity, expressed as a multiple of the value at a low reference RH (typically <40%). g(RH) is analogous to f(RH) but describes the change in size of particles as a function of RH |
| PNSD[1] | The **particle number size distribution** describes the number of particles in multiple specified size ranges. The PNSD can provide information about formation processes such as new particle formation, aerosol transport as well as aerosol types. |





| CN, CCN, IN | The **particle number concentration (CN)** refers to the number of particles per unit volume of air ($cm^{-3}$). The **Cloud Condensation Nuclei (CCN)** number concentration is the number of aerosol particles which can activate to a cloud droplet at a given supersaturations of water. The **Ice Nuclei (IN)** is the number of aerosol particles onto which water freezes following various processes. CCN is often indicated as a percent of the total CN for specific supersaturation typical of atmospheric cloud formation. CCN number concentration is sometimes approximated using the fraction of particles larger than a given diameter from the particle number size distribution |
|---|---|
| $Fz(\sigma_{ep})^{1,2}$ | The profile of the **particle light extinction coefficient** is the spectrally dependent sum of aerosol particle light scattering and absorption coefficients per unit of geometrical path length. |
| Aerosol chemical composition[*] | The chemical composition of aerosol particles is often expressed in $ug\ m^{-3}$. For climate applications, only the main components of the aerosol composition are relevant, i.e. influencing the aerosol hygroscopic properties and refractive index. Total inorganic, **Elemental Carbon (EC)** and **Organic Carbon (OC)** mass concentrations are, in a first approximation, sufficient. |

**Table 1:** Measured and derived aerosol particle properties relevant to radiative forcing on climate (adapted from GAW Report 227). [1]Variables currently recognized as core aerosol variables by WMO/GAW [2]Variables currently recognized as ECVs for Global Climate Monitoring application areas (GCOS).



| Aerosol variable | Instrument used | Time resolution (raw) | Associated Uncertainty |
|---|---|---|---|
| particle light scattering coefficient ($\sigma_{ep}$) | Integrating Nephelometer 3563 (TSI Inc, USA); Aurora 3000 (Ecotech Inc, AU); NGN-2 (Optec Inc, USA); Aurora M9003 (Ecotech Inc, AU) | 1 min | 10% (from Sherman et al., 2015, extended to other nephelometers) |
| particle light absorption coefficient ($\sigma_{ap}$) | Multi Angle Absorption Photometer model 5012 (MAAP, by THERMO-Scientific Inc. USA); Continuous Light Absorption Photometer (CLAP, NOAA); Aethalometer (AE16, AE31, AE33) (Magee Scientific, USA). Particle/Soot Absorption Photometer (PSAP, Radiance Research Inc) | 1 min | 20% (from Sherman et al., 2015, extended to other filter-based photometers) |
| Particle Number concentration (CN) | CPC & MPSS | 1 min (CPC) to 5 min (MPSS) | 10% for particles >15 nm (from Wiedensohler et al., 2012) |

**Table 2:** Instruments used for the determination of aerosol optical and physical properties in SARGAN, original time resolution for raw data and associated uncertainties.





| Station Name | GAW Code | Country/ Region | GPS coordinates | Site Characteristics | $\sigma_{sp}$ starting year | $\sigma_{ap}$ starting year | PNSD starting year | CN starting year |
|---|---|---|---|---|---|---|---|---|
| **WMO I, Africa** | | | | | | | | |
| Cape Point | CPT | ZA | 34°21'S,18° 29'E,230m | Coast, RB | 2005 GA,N | 2005 GA,N | - | 2005 GA,N |
| Izana | IZO | ES | 28°18'N,16°29'W,2373m | Mt, Mix | 2008 A,GA | 2006 A,GA | 2008 GA | 2006 GA |
| La Réunion - Maïdo atmospheric observatory | RUN | FR | 21°4'S,55° 22'E,2160m | Mt, Mix | - | 2014 A,GA | 2016 A,GA | - |
| Welgegund | WGG | ZA | 26°34'S,26° 56E,1480m | Con, U | - | - | 2010 NOT | - |
| **WMO II, Asia** | | | | | | | | |
| Anmyeon –do | AMY | KR | 36°32'N,126° 19'E,46m | Coast, RB | 2008 GA | 2008 GA | 2017 GA | - |
| Gosan | GSN | KR | 33°16'N,126° 10'E,72m | Coast, RB | 2001 GA,N | 2001 GA,N | - | 2008 GA,N |
| Lulin | LLN | TW | 23°28'N, 120° 52'E,2862m | Mt, F | 2008 GA,N | 2008 GA,N | - | 2009 GA,N |
| Mukstewar | MUK | IN | 29°26'N,79°37'E,2180m | Mt, Mix | 2006 NOT | 2006 NOT | - | - |
| Pha Din | PDI | VN | 21°34'N, 103° 30'E,1466m | Mt, F | 2008 GA,N | 2008 GA,N | - | - |
| Mt. Waliguan | WLG | CN | 36°17'N,100° 54'E,3810m | Mt, Mix | 2005 GA,N,NOT | 2005 GA,N,NOT | - | 2005 GA,N,NOT |
| **WMO III, South America** | | | | | | | | |
| Mount Chacaltaya | CHC | BO | 16°12'S,68°5'W,5320m | Mt, Mix | 2012 A,GA | 2011 A,GA,GU | 2012 A,GA,GU | - |
| El Tololo | TLL | CL | 30° 10'S,70°47'W,2220m | Mt, Mix | 2013 GA | 2016 GA | - | - |
| **WMO IV, North America, Central America and the Caribbean** | | | | | | | | |
| Acadia National Park-McFarland Hill | ACA | US | 44° 22'N,68° 15'W, 150m | Coast, RB | 1993 GA,I | - | - | - |
| Alert | ALT | CA | 82° 29'N,62° 20'W,210 m | Polar, Coast, P | 2004 GA,N | 2004 GA,N | - | 2004 GA,N |
| Appalachian State University, Boone | APP | US | 36° 12'N,81° 42'W,1100m | Con, RB | 2009 GA,N | 2009 GA,N | - | 2009 GA,N |
| Big Bend National Park-K-Bar | BBE | US | 29°18'N,103° 10'W,1056m | Con, DE | 1998 GA,I | - | - | - |
| Bondville | BND | US | 40° 2'N,88° 22'W,213m | Con, RB | 1994 GA,N | 1996 GA,N | - | 1994 GA,N |
| Barrow | BRW | US | 71° 19'N, 156° 36'W, 11m | Polar, Coast, P | 1993 GA,N | 1991 GA,N | - | 1990 GA,N |
| Cape San Juan | CPR | PR | 18° 22'N,65° 37'W,65m | Coast,F | 2004 GA,N | 2006 GA,N | - | 2004 GA,N |
| Columbia River Gorge | CRG | US | 45° 39'N,121° 0'W,178m | Con, RB | 1993 GA,I | - | - | - |
| Egbert | EGB | CA | 44°13'N,79° 47'W,255m | Con, RB | 2009 GA,N | 2009 GA,N | - | 2011 GA,N |
| East Trout Lake | ETL | CA | 54° 21'N,104° 59'W,500 m | Con, F | 2008 GA,N | 2008 GA,N | - | 2008 GA,N |
| Great Basin National Park-Lehman Caves | GBN | US | 39° 0'N,114° 12'W,2067m | Mt, DE | 2007 GA,I | - | - | - |
| Glacier National Park-Fire Weather Station | GLR | US | 48° 30'N,113° 59'W,980m | Con, F | 2007 GA,I | - | - | - |




| Station Name | GAW Code | Country/Region | GPS coordinates | Site Characteristics | σsp starting year | σap starting year | PNSD starting year | CN starting year |
|---|---|---|---|---|---|---|---|---|
| Great Smoky Mountains NP | GSM | US | 35° 38'N,83° 56'W,810m | Con, F | 1993[GA,I] | - | - | - |
| Grand Teton National Park | GTT | US | 43°40'N,110° 36'W,2105m | Con, F | 2011[I] | - | - | - |
| Hance Camp at Grand Canyon NP | HGC | US | 35°58'N,111°59'W,2267m | Con, F | 1997[GA,I] | - | - | - |
| Mammoth Cave National Park-Houchin | MCN | US | 37° 7'N,86°8'W,236m | Con, RB | 1993[GA,I] | - | - | - |
| Mount Rainier National Park-Tahoma Woods Meadow | MRN | US | 46° 45'N,122° 7'W, 424m | Con, F | 1993[GA,I] | - | - | - |
| Mount Zirkel Wilderness | MZW | US | 40° 32N ,106° 40'W,3243m | Mt, F | 1993[GA,I] | - | - | - |
| National Capitol - Central, Washington D.C | NCC | US | 38° 53'N,77° 2'W,514m | Con, U | 2003[GA,I] | - | - | - |
| Phoenix | PAZ | US | 33°30'N,112°5'W,342m | Con,U | 1997[GA,I] | - | - | - |
| Rocky Mountain NP | RMN | US | 40°16'N ,105°32'W,2760m | Mt, RB | 2008[GA,I] | - | - | - |
| Sycamore Canyon | SCN | US | 35° 8'N,111° 58'W,2046m | Con F | 1998[GA,I] | - | - | - |
| Southern Great Plains E13 | SGP | US | 36° 36'N,97° 29'W,318m | Con, RB | 1995[GA,N] | 1996[GA,N] | - | 1996[GA,N] |
| Shenandoah National Park-Big Meadows | SHN | US | 38° 31'N, 78° 26'W,1074m | Con, F | 1996[GA,I] | - | - | - |
| Steamboat Springs Colorado (Storm Peak Lab.) | SPL | US | 40°26'N,106° 44'W,3220m | Mt, F | 2011[GA,N] | 2011[GA,N] | - | 1998[GA,N] |
| Trinidad Head | THD | US | 41° 3'N,124° 9'W,107m | Coast, RB | 2002[GA,N] | 2002[GA,N] | - | 2002[GA,N] |
| **WMO V, South-West Pacific** | | | | | | | | |
| Bukit Kototabang | BKT | ID | 0° 12S,100° 19E,864m | Mt, F | - | 2015[GA] | - | - |
| Cape Grim | CGO | AU | 40° 40S,144° 41E,94m | Coast, RB | 2012[GA] | 2011[GA] | - | 2013[GA] |
| Mauna Loa | MLO | US | 19°32'N,155° 34W,3397m | Mt, Mix | 1974[GA,N] | 2000[GA,N] | - | 1974[GA,N] |
| Samoa (Cape Matatula) | SMO | US | 14° 14S,170° 33E,77m | Coast, P | - | - | - | 1977[GA,N] |
| **WMO VI, Europe** | | | | | | | | |
| Annaberg-Buchholz | ANB | DE | 50° 34'N,12° 59'E,545m | Con, U | - | 2012[GA,GU] | 2012[A,GU] | - |
| Aspvreten | APT | SE | 58°47'N,17°229'E, 20m | Coast, RB | - | 2008[A,E,GA] | 2005[A,E,GA] | - |
| El Arenosillo | ARN | ES | 37° 6'N, 6° 43'W,41m | Coast, F | 2005[A,GA,N] | 2012[A,GA,N] | 2016[A,GA] | 2017[A,GA] |
| Birkenes II | BIR | NO | 58° 23'N, 8° 15'E,219m | Con, F | 2009[A,E,GA] | 2009[A,E,GA] | 2009[A,E,GA] | - |
| BEO Moussala | BEO | BG | 42° 10'N,23° 34'E,2971m | Mt, Mix | 2007[A,GA,N] | 2012[A,GA,N] | 2008[A,GA] | - |
| Mt Cimone | CMN | IT | 44°10'N,10 ° 41'E,2165m | Mt, Mix | 2007[A,GA] | 2011[A,GA] | 2006[GA] | 2008[A,GA] |
| DEM_Athens | DEM | GR | 37° 59'N,23° 48'E,270m | Coast, U | 2012[A,GA] | 2012[A,GA] | 2015[A,GA] | - |
| Dresden-Nord | DRN | DE | 51° 3'N,13° 44'E,116m | Con, U | - | - | 2001[GU] | - |



| Station Name | GAW Code | Country/Region | GPS coordinates | Site Characteristics | $\sigma_{sp}$ starting year | $\sigma_{ap}$ starting year | $\sigma_{sp}$ starting year | PNSD starting year | CN starting year |
|---|---|---|---|---|---|---|---|---|---|
| Dresden-Winckelmannstrasse | DRW | DE | 51° 2'N,13° 43'E, 120m | Con, U | - | | | 2010 [GU] | - |
| Deutschneudorf | DTC | DE | 50° 36'N,13° 27'E,660m | Con, U | | | | 2017 [A] | - |
| Lecce (University of Salento) | ECO | IT | 40° 20'N, 18° 6'E,30m | Coast, F | 2015 [A,GA] | | | - | - |
| Finokalia | FKL | GR | 35°19'N,25° 40'E,250m | Coast, RB | 2004 [A,GA] | 2000 [A,GA] | | 2009 [A,GA] | 2006 [EGA] |
| SIRTA Atmospheric Research Obs. | GIF | FR | 48° 42'N,2° 9'E,162m | Con, U | 2012 [A,GA] | 2010 [A,GA] | | 2017 [A,GA] | - |
| Helmos Mountain | HAC | GR | 37° 59'N,22° 11'E,2340m | Mt, Mix | 2016 [A,GA] | 2016 [A,GA] | | 2016 [A,GA] | - |
| Hohenpeissenberg | HPB | DE | 47° 48'N, 11°0'E,985m | Mt, RB | 2006 [A,E,GA] | 2004 [A,GA] | | 1998 [A,GA,GU] | 1995 [A,E,GA] |
| Hyytiälä | HYY | FI | 61° 51'N,24° 16'E,181m | Con, F | 2006 [A,GA] | 2006 [A,GA] | | 1996 [A,GA] | 2005 [A,GA] |
| Ispra | IPR | IT | 45°47'N,8° 37'E,209m | Con, RB | 2004 [A,E,GA] | 2004 [A,E,GA] | | 2008 [A,E,GA] | - |
| Jungfraujoch | JFJ | CH | 46° 32'N,7° 59'E,3578m | Mt, Mix | 1995 [A,E,GA] | 2001 [A,E,GA] | | 1997 [A,E,GA] | 1995 [A,E,GA] |
| Kosetice | KOS | CZ | 49° 34'N,15° 4'E,535m | Con, RB | 2012 [A,E,GA] | 2012 [A,E,GA] | | 2008 [A,E,GA] | - |
| K-puszta | KPS | HU | 46° 58'N,19°34'E,125m | Con, RB | 2006 [A,E,GA,N] | 2006 [A,E,GA,N] | | 2006 [A,GA] | - |
| Leipzig TROPOS | LEI | DE | 51°21'N,12° 26'E,113m | Con, U | | 2009 [A,GU] | | 2010 [A,GA,GU] | - |
| Leipzig-Eisenbahnstrasse | LEI-E | DE | 51° 20'N,12° 24'E,120m | Con, U | | 2009 [A,E,GU] | | 2011 [A,GA,GU] | - |
| Leipzig-Mitte | LEI-M | DE | 51° 20'N,12° 22'E,111m | Con, U | | 2010 [A,E,GU] | | 2010 [A,GU] | - |
| Madrid | MAD | ES | 40° 27'N,3° 43'W,669m | Con, U | | | | 2014 [A] | - |
| Melpitz | MEL | DE | 51° 31'N,12° 56'E,86m | Con, RB | 2007 [A,GA,GU] | 2007 [A,E,GA,GU] | | 1995 [GA,GU] | - |
| Montsec | MSA | ES | 42°3'N,0°43'E,1571m | Mt, Mix | 2015 [A,GA,N] | 2013 [A,GA,N] | | - | 2016 [A,GA] |
| Montseny | MSY | ES | 41° 46'N,2° 21'E,700m | Mt, Mix | 2010 [A,GA,N] | 2009 [A,GA,N] | | 2009 [A,GA] | 2013 [A,GA] |
| Neuglobsow | NGL | DE | 53° 10'N,13° 1'E,62m | Con, F | | 2015 [A,GA,GU] | | 2011 [A,GA,GU] | - |
| Obs. Perenne de l'Environnement | OPE | FR | 48° 33'N,5° 30'E,392m | Con, RB | 2012 [A,GA] | 2012 [A,GA] | | 2016 [A,GA] | - |
| Pallas (Sammaltunturi) | PAL | FI | 67° 58'N,24° 6'E,565m | P, RB | 2000 [A,GA] | 2007 [A,GA] | | 2000 [A,E,GA] | 1996 [A,E,GA] |
| Payerne | PAY | CH | 46° 48'N,6° 56'E,489m | Con, RB | | 2015 [E] | | - | - |
| Pic du Midi | PDM | FR | 42° 56'N,0° 8'E,2877m | Mt, Mix | 2016 [A,GA] | 2013 [A,GA] | | - | 2017 [A,GA] |
| Prague-Suchdol | PRG | CZ | 50° 7'N,14° 23'E,270m | Con, U | | | | 2012 [A,E,GA] | - |
| Puy de Dome | PUY | FR | 45° 46'N,2° 57'E,1465m | Mt, Mix | 2006 [A,GA] | 2008 [A,E,GA] | | 2007 [A,E,GA] | 2005 [A,GA] |
| Rigi | RIG | CH | 47° 4'N,8° 27'E,1031m | Coast, RB | | 2015 [E] | | - | - |
| Sonnblick | SNB | AT | 47° 3'N,12° 57'E,3106 | Mt, Mix | | | | - | 2014 [A,E] |
| Schauinsland | SSL | DE | 47° 54'N,7° 54'E, 1205m | Con, F | | 2009 [A,GA,GA,GU,E] | | 2005 [A,GA,GU] | - |
| Summit | SUM | DK | 72° 34'N,38° 28'W,3238m | Polar, P | 2011 [GA,N] | 2003 [GA,N] | | - | - |
| Tiksi | TIK | RU | 71° 35'N,128° 55'E,8 m | P, Coast, RB | 2015 [A,GA] | 2007 [A,GA] | | - | - |





| Station Name | GAW Code | Country/ Region | GPS coordinates | Site Characteristics | $\sigma_{sp}$ starting year | $\sigma_{ap}$ starting year | PNSD starting year | CN starting year |
|---|---|---|---|---|---|---|---|---|
| Granada | UGR | ES | 37° 9'N,3° 36'W,680m | Con, U | 2006 [A,GA,N] | 2006 [A,GA,N] | - | - |
| Värriö | VAR | FI | 67° 46'N,29° 34'E,400m | P, RB | - | - | 2000 [GA] | 1992 [AGA] |
| Vavihill | VAV | SE | 56°1'N,13° 9'E, 175m | Con, F | 2008 [A,E,GA] | 2008 [A,E,GA] | 2001 [A,E,GA] | - |
| Waldhof | WAL | DE | 52° 48'N,10° 45'E,74m | Con, F | - | 2013 [A,GA,GU] | 2009 [A,GA,GU] | - |
| Zeppelin mountain | ZEP | NO | 78° 54'N, 11° 53'E,474m | P, Mt, P | 2008 [A,GA] | 2002 [AGA] | 2000 [A,E,GA] | 2010 [E,GA] |
| Zugspitze-Schneefernerhaus | ZSF | DE | 47° 24'N,10° 58'E,2671m | Mt, Mix | 2010 [A,E,GA] | 2009 [A,GA,GU] | 2004 [CA,CU] | - |
| **WMO VII, Antarctica** | | | | | | | | |
| Neumayer | NMY | DE | 70° 39'S ,8°15'W,42m | P, Coast, Mix | 2001 [E,GA] | 2006 [GA] | - | 1995 [E,GA] |
| South Pole | SPO | US | 89° 59'S,24° 47'W,2841m | P, P | 1979 [GA,N] | 2017 [GA,N] | - | 1974 [GA,N] |
| Trollhaugen | TRL | NO | 72° 0'S,2° 32'E,1553m | P, P | 2014 [A,E,GA] | 2014 [A,E,GA] | 2014 [A,E,GA] | - |

**Table 3:** list of sites in the SARGAN network in 2017 or last year with data in EBAS used in the present study. Table indicates the starting year for each variable, and the site geographical category: Mountain=Mt, Polar=P, Continental=Con, Coastal =Coast, and the air mass footprint characteristics: Rural background=RB, Forest=F, Desert=DE, (Sub-)Urban=U, Pristine= P Mixed: Mix, ACTRIS=A, EMEP=E, GAW-WDCA=GA, GUAN=GU, IMPROVE=I, NOAA- FAN=N, Not in EBAS=NOT. Sites highlighted in grey closed in 2017 or earlier.





| GCOS Principles | SARGAN Response to GCOS Principles |
|---|---|
| The impact of new systems or changes to existing systems should be assessed prior to implementation. | All instruments used in SARGAN should be accepted in the standard procedures. Whenever instruments are custom-made or modified from commercial versions (e.g., SMPS), they must be intercompared with a reference instrument operated by a calibration center. |
| A suitable period of overlap for new and old observing systems should be required. | While this was not necessarily implemented in the past, it is now the case that any upgrade in the instrumental deployment at a SARGAN site should be made by maintaining side-by-side measurements with the old and new instruments for an extended measurement period. |
| The results of calibration, validation and data homogeneity assessments, and assessments of algorithm changes, should be treated with the same care as data. | All results from intercomparison exercises are made public and should be conserved by the Calibration Centers. |
| A capacity to routinely assess the quality and homogeneity of data on extreme events, including high-resolution data and related descriptive information, should be ensured. | Within the contributing networks to SARGAN, tools for online quality control of instrument performance are used to ensure data quality. All information on data quality is traceable, including availability of raw information, conserved by the data centers. RAW information (level 0) is available for reprocessing in case it is required for analyzing specific events |
| Consideration of environmental climate-monitoring products and assessments, such as IPCC assessments, should be integrated into national, regional and global observing priorities. | SARGAN supports the implementation of UNFCCC policy-driven networks established to respond to EU-directive (local and European air-quality networks), to the Convention on Long Range Trans-boundary Air Pollution (CLRTAP) of the United Nations Economic Commission for Europe (UNECE) contribution to WMO's Global Climate Observing System (GCOS) |
| Uninterrupted station operations and observing systems should be maintained. | The analyses of SARGAN data coverage shows that the network is composed of stations that are, for the most part, providing continuous data; some sites have been doing so for decades. |
| A high priority should be given to additional observations in data-poor regions and regions sensitive to change. | While we acknowledge that the situation is still not satisfactory, a number of stations have been implemented in the framework of GAW in the last decade or so and have improved availability of data from regions where coverage was, previously, totally lacking. |
| Long-term requirements should be specified to network designers, operators and instrument engineers at the outset of new system design and implementation. | Almost all stations are registered to GAW as a regional, global or contributing station and are documented in the GAWSIS metadata base. |
| The carefully-planned conversion of research observing systems to long-term operations should be promoted. | This work is supported by the establishment of relevant European Research Infrastructures or networks that are clearly established in the long-term with commitments at country ministerial levels |
| Data management systems that facilitate access, use and interpretation should be included as essential elements of climate monitoring systems. | Considerable work has been carried out in recent years to facilitate access to all SARGAN information through the development of tools in WDCA to facilitate uptake and accessibility of information. |

**Table 4:** a description of the status of SARGAN with respect to the requirements for GCOS networks




|  |  | Threshold | Breakthrough | Goal |
|---|---|---|---|---|
| Resolution | Spatial Resolution: Horizontal All SARGAN variables | 1000 km | 500 km | 100 km |
|  | Spatial Resolution: Vertical All SARGAN variables | Not applicable | Not applicable | Not applicable |
|  | Temporal Resolution All SARGAN variables | 1 month | 1 day | 1 hour |
|  | Timeliness All SARGAN variables | annual | 24h-delay | 6h-delay |
| Uncertainty | Required measurement uncertainty $\sigma_{sp}$ $\sigma_{ap}$ CN $\omega_0$ | 10% 20% 10% 20% | 10% 20% 10% 20% | 5% 10% 5% 10% |
|  | Stability for users $\sigma_{sp}$ and $\sigma_{ap}$, CN, $\omega_0$ | 2 %/yr | 1%/yr | 0,5%/yr |

**Table 5:** proposed requirements for GCOS application area for SARGAN variables






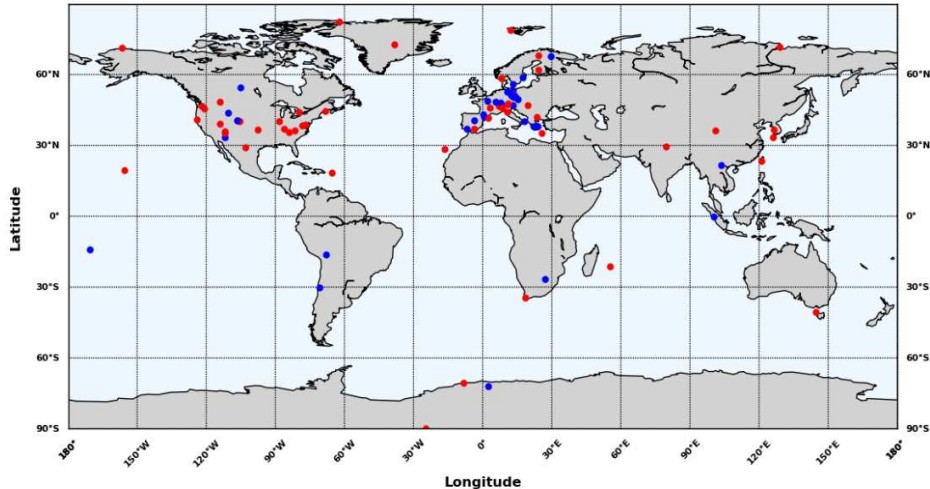

**Figure 1:** Location of sites contributing to the present study. In blue, sites which provided information for the reference year 2017 and in red, sites that in addition, provided >10year time series for optical properties used in Collaud Coen et al. (submitted).






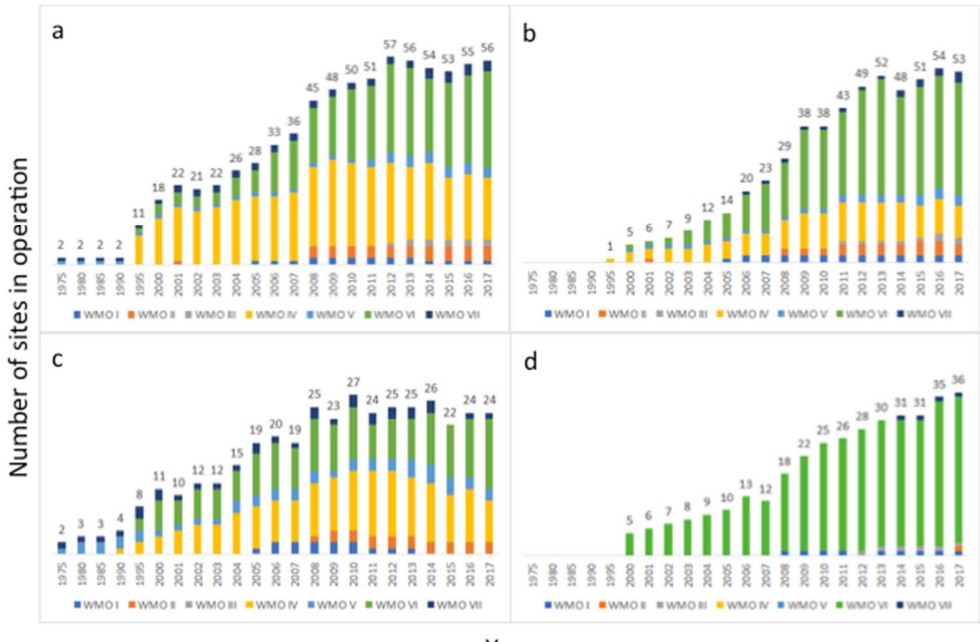

**Figure 2:** The evolution of data provision in SARGAN for optical (panel a, b), and physical parameters split between sites providing particle number concentration (panel c) and sites providing Particle Number size distribution (panel d) over the period 1975-2017 for the WMO regions. WMO I = Africa, WMO II = Asia, WMO III = South America, WMO IV = North America, Central America and the Caribbean, WMO V = the South-West Pacific, WMO VI = Europe, WMO VII = Polar




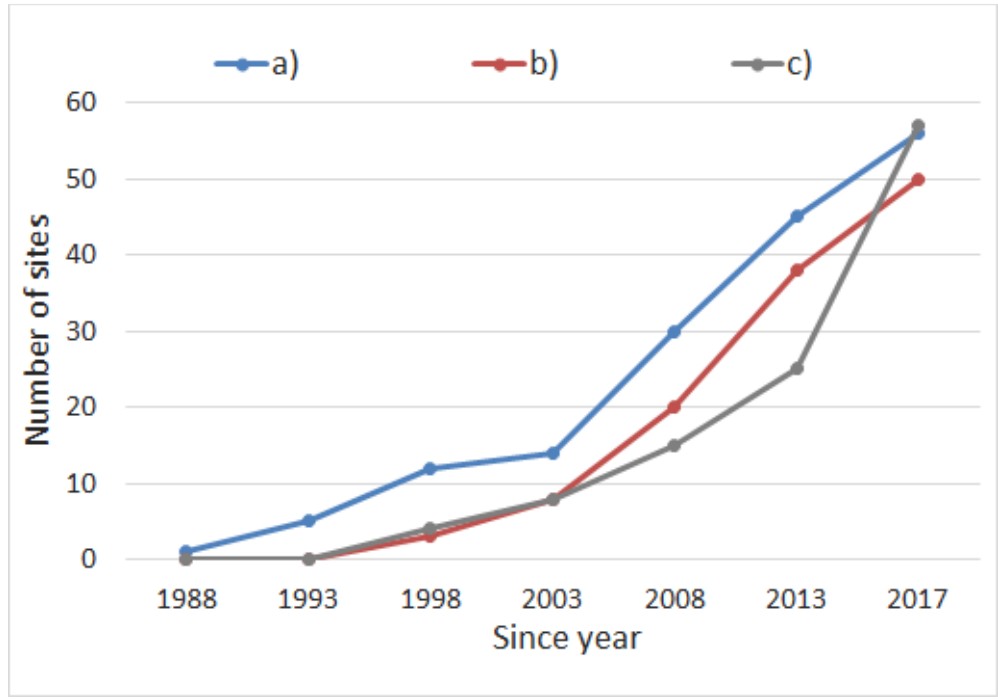

**Figure 3:** Cumulative number of sites providing information to WDCA for the aerosol variables: a) scattering, b) absorption and c) combined size and particle number concentration.





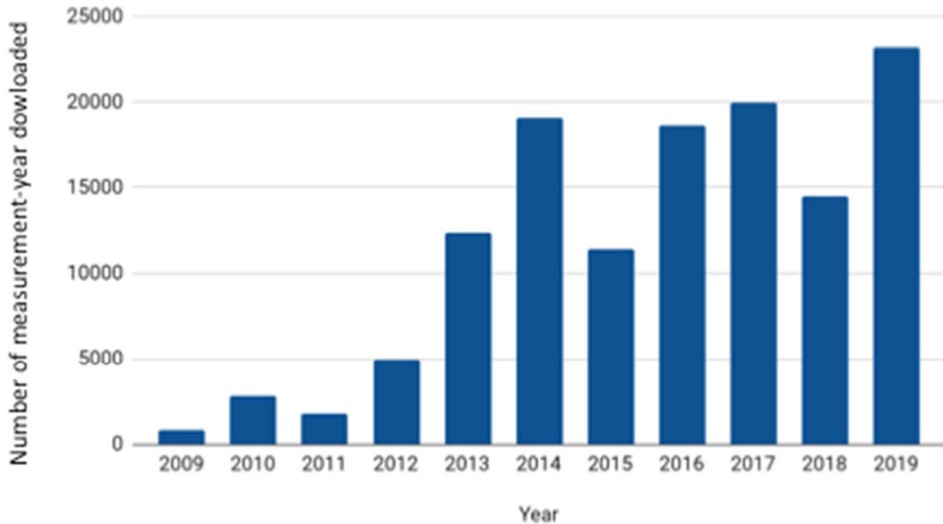

**Figure 4:** The use of SARGAN data from GAW-WDCA over the period May 2009 - October 2019 as indicated by the number of full years of measurement data downloaded each year. Data extracts as tailored special delivery (the full data base for a special purpose) are not included




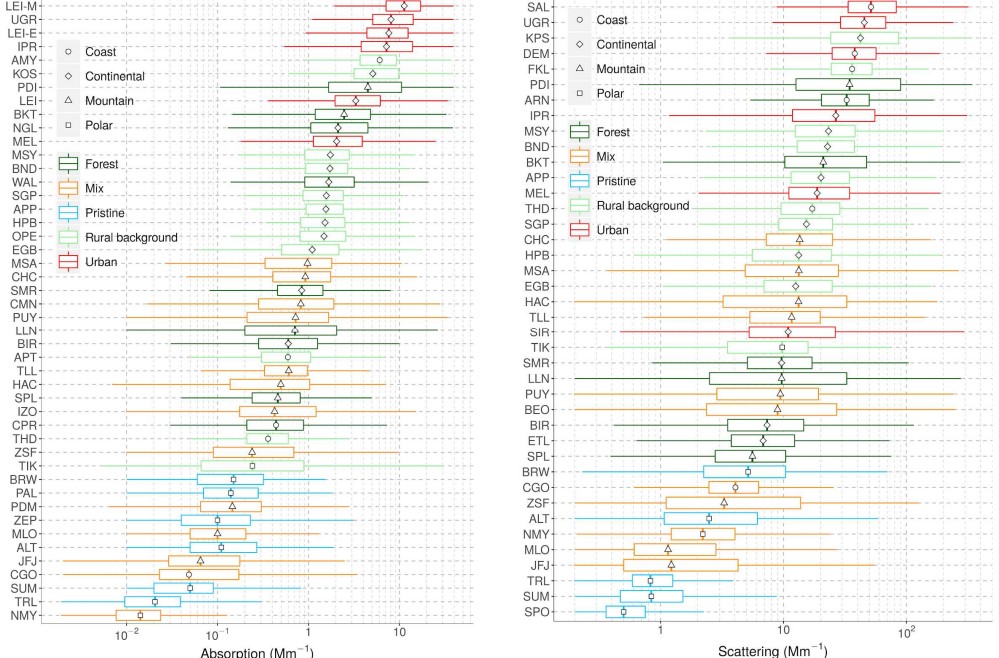

**Figure 5.** Boxplot of hourly aerosol absorption (left panel) and scattering (right panel) coefficients at the SARGAN sites with sufficient annual coverage over the considered period (see summary Tables SM1 and SM2 for details). Boxplot colour indicates the footprint, the symbol at the median indicates the geographical category; both colour and symbol follow Table 3.





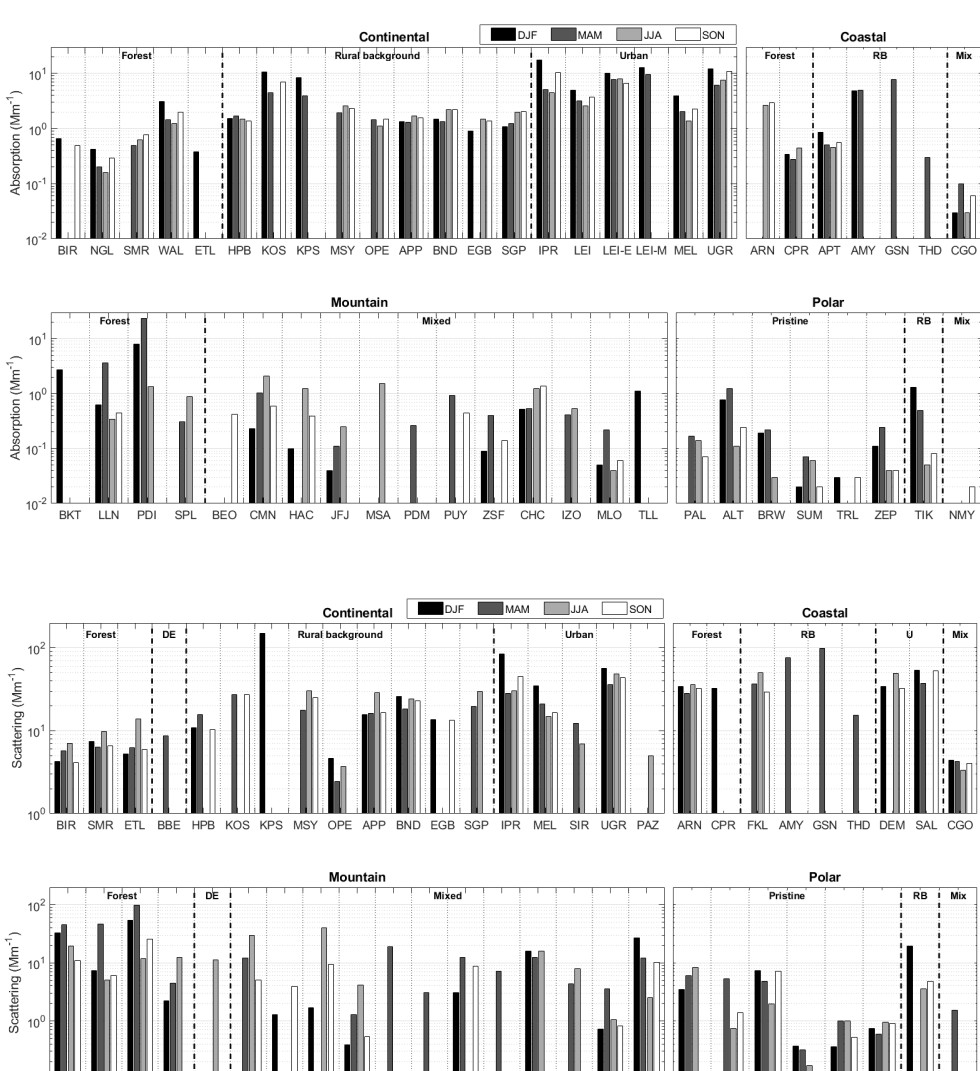

**Figure 6:** variation of the seasonal median of absorption coefficient (upper panel) and scattering coefficient (lower panel) for the different sites, grouped according to site classification of Table 3




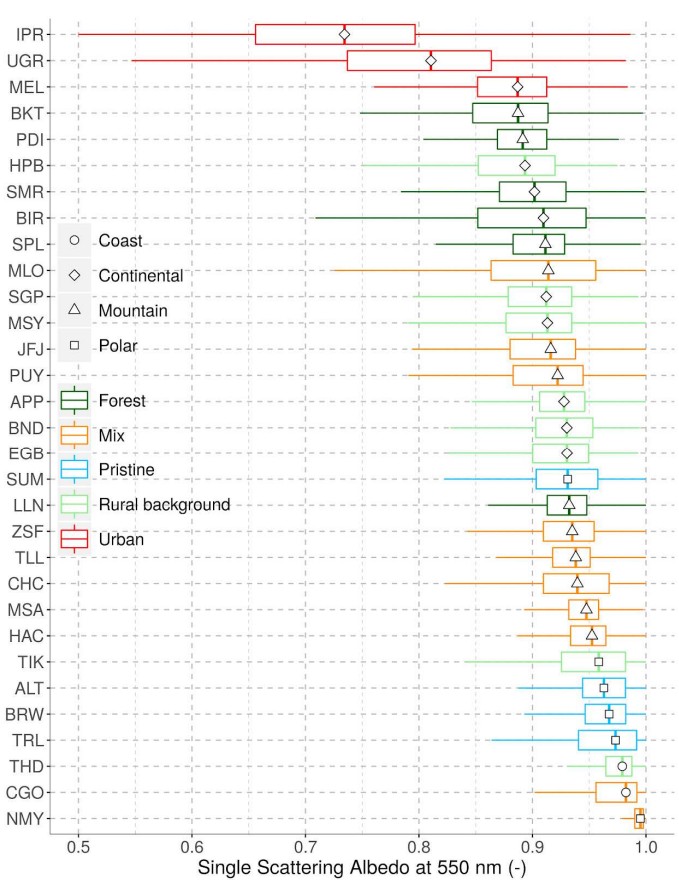

**Figure 7:** Boxplot of quantiles and annual median for single scattering albedo at 550 nm at the analyzed sites. Box color indicates the footprint according to table 3.

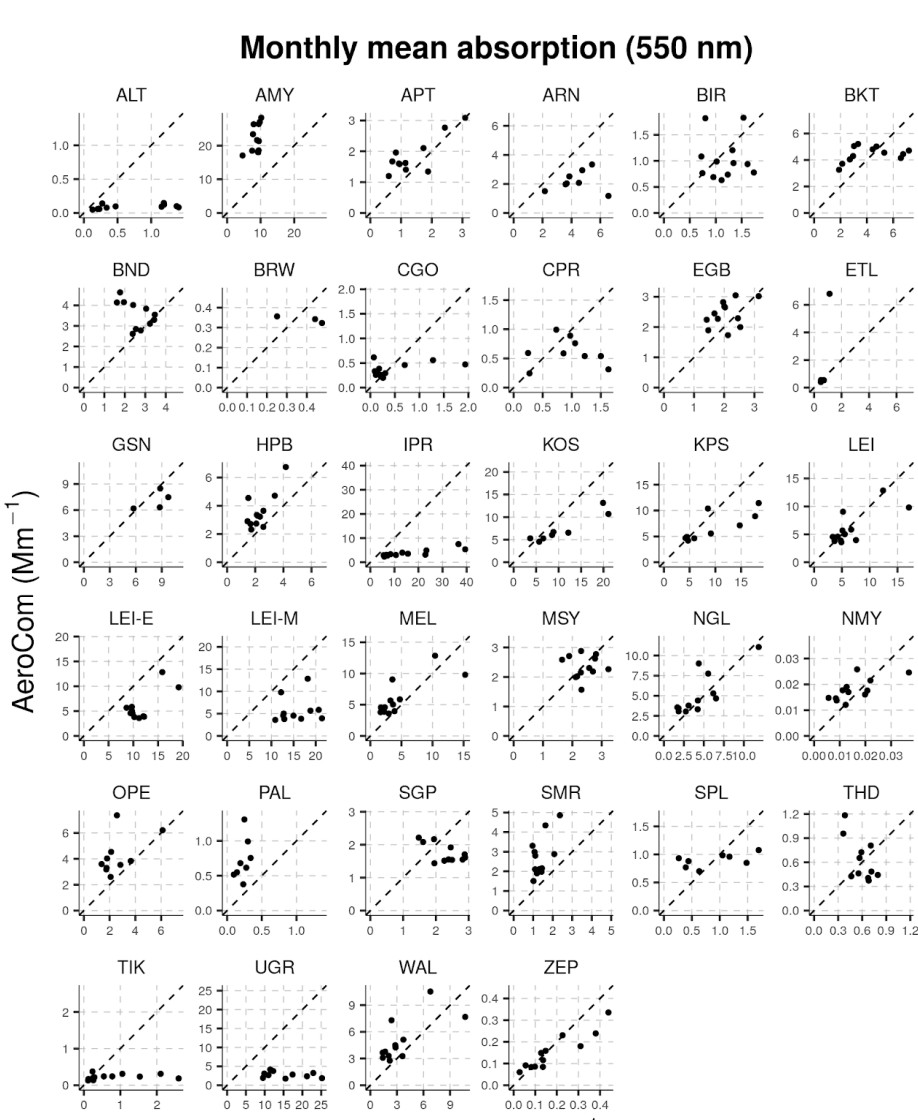

**Figure 8a:** AeroCom-SARGAN comparison of seasonal means of absorption coefficients at selected surface sites.


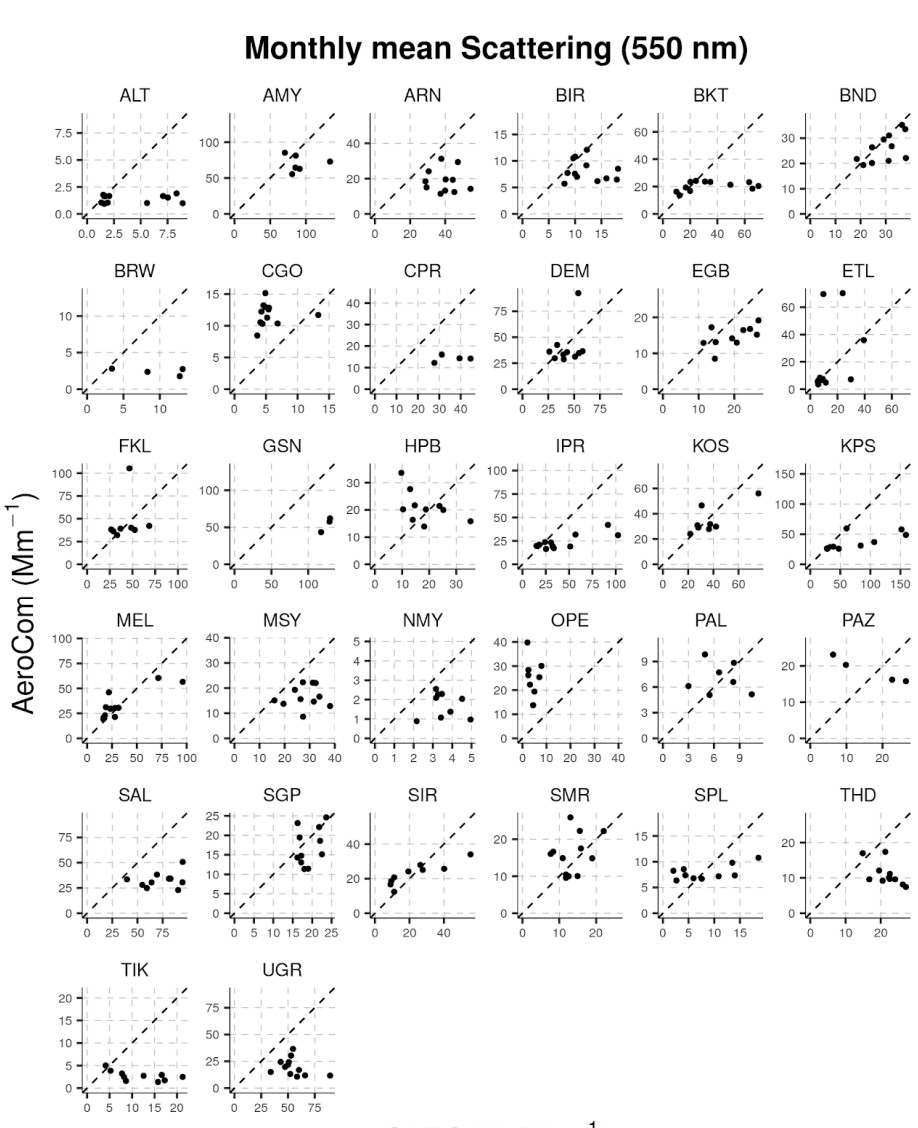

**Figure 8b:** AeroCom-SARGAN comparison of seasonal means of scattering coefficients at selected surface sites.





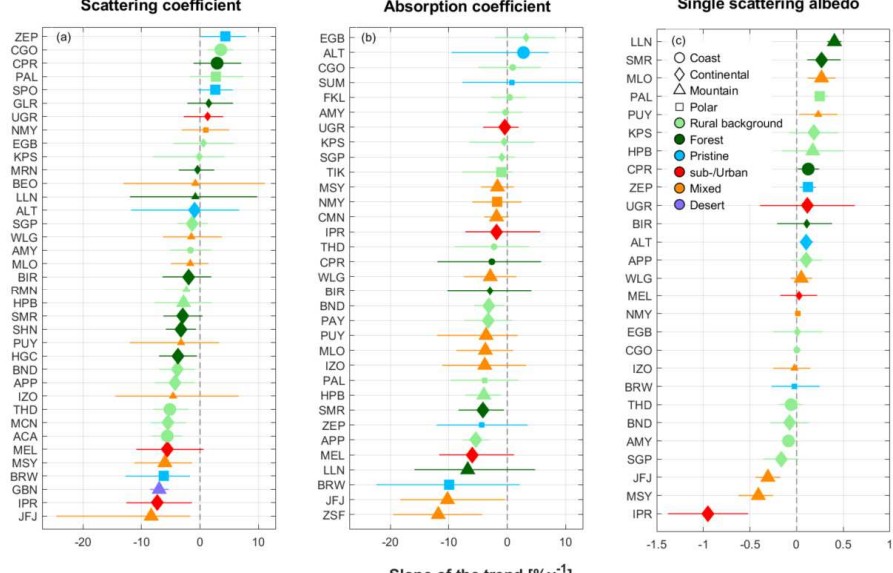

**Figure 9:** Annual trends for (a) the scattering coefficient, (b) the absorption coefficient and (c) the single scattering albedo derived for SARGAN stations providing 10yr time series ending in 2016-2018. The larger symbols represent statistically significant trends at 95% significance level derived from the Mann-Kendall seasonal test. The lines are the 90% confidence limit upper and lower confidence limits derived from the Sen's slope estimator.





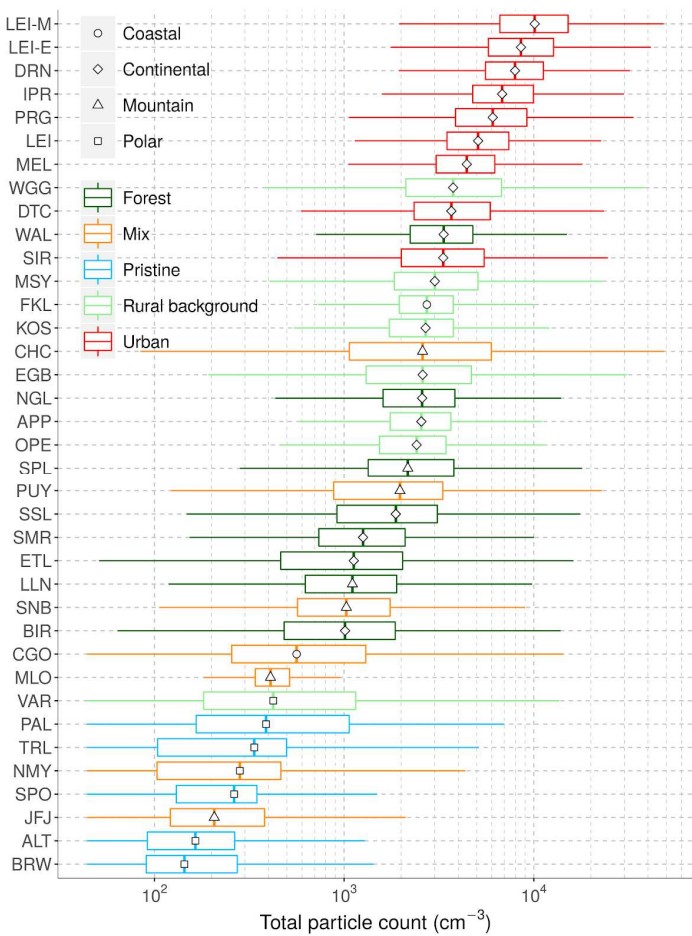

**Figure 10 :** Yearly median of the total particle number concentration ($N_{tot}$). The markers represent the median of the data and the lower and upper edges of the box indicate the 25th and 75th percentiles, respectively. The length of the whiskers represents 1.5 interquartile range. Different markers and box colors indicate geographical categories and footprint, respectively, according to Table 3. Note that only the sites with sufficient annual coverage (i.e. >75%) are presented






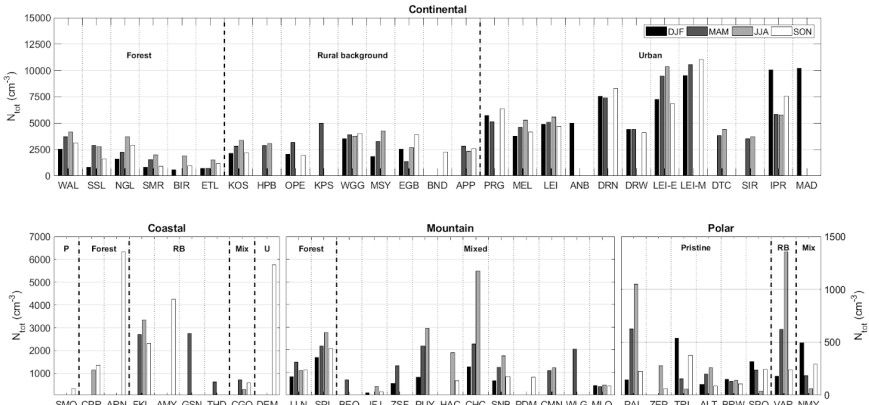

**Figure 11:** Seasonal medians of the total particle number concentration ($N_{tot}$). Stations are grouped according to their geographical category, and are further sorted based on their dominant footprint. For each site/season, statistics are only presented when corresponding data availability is >75%. Note that the same scale is used for coastal and mountain sites.