# Peer review of "A global analysis of climate-relevant aerosol properties retrieved from the network of GAW near-surface observatories"

_Atmospheric Measurement Techniques, 2019_

## Referee Comment (RC1) · Anonymous Referee #1 · 6 Mar 2020

The results of a large effort to summarize the climate-relevant in situ aerosol properties available at all sites connected to the GAW network are presented. The growth of the global base over the past few decades is shown as well as the increase in the usage of the data over the past ten years. Values of absorption, scattering, and SSA are compared for the wide range of existing sites. Seasonal data and annual trends are also presented. All in all, it is a substantial effort that is definitely worthy of publication in AMT. Relatively minor comments are listed below.

Table 1. Should the description of CCN say "CCN number concentration is sometimes approximated using the fraction of particles larger than a given diameter from the par-

[Figure]

ticle number size distribution AND AN ASSUMED CHEMICAL COMPOSITION"?

Line 225: ". . .but also including sites IN other WMO regions. . ."

Line 249: "..many of them ARE no longer documented. . ."

Line 271: Analyzes instead of analyses?

Line 271: Define SARGAN here where it is first used.

Lines 320 – 325: It would be handy to have a reference to the relevant report of the GAW measurement guidelines included in Table 2 , i.e. WMO/GAW Report #200 for light scattering and absorption.

Line 364: define the ultrafine and fine ranges in terms of diameter.

Lines 400 – 413: When was this change made, i.e. only removing data affected by instrument issues or contamination? Were older data sets amended?

Lines 439 – 440: Are there references for the "manual-expansion type" and "automated version" that can be provided so the reader knows what these particle counters are?

Line 484: define kerbside.

Line 540: A couple of sentences with references on different methods that have been used to calculate trends would be helpful here.

Line 551: ". . ..too high, however, OBSERVATIONS INDICATE it is. . .."

Lines 551 – 552: What exactly is meant by "While the bias values are robust at the sites investigated. . ."? The bias values (i.e. model – measurement differences) are well characterized or low?

Lines 565 – 569: How did the measured and modelled number size distributions compare?

Line 570: ". . .is the evaluation OF MEASURED AND MODELLED cloud condensation

nuclei."

Line 580: Provide the link to GAWSIS here where it is first mentioned.

Line 587 – 589: What is the connection between not all GAW stations being able to measure all variables listed in Table 1 and SARGAN being a subset of stations in GAW? Please clarify.

Line 628: Where are WMO regions I, II, III, and IV? I don't think this is stated previously.

Line 634: What is meant by "a station footprint that is large"? Is this related to its representativeness of a region? Or land type?

Line 735: "...for 29 of these sites IT was possible..."

Lines 744 – 746: Is it possible to cite a reference for this assumption?

Line 772: It is more commonly thought that Cape Grim is a coastal Southern site, than a Pacific site.

Table 3: Abbreviations shown in the plots (DE, RB, U) should be defined in the caption.

Figure 8a: The title says monthly means while the caption says seasonal means. Given the number of points, I assume it is the former.

Figure 8: Coloring the points by month may provide useful information on under- and overestimates by the models. Also – 8a shows absorption and 8b shows scattering but scattering is introduced in the text first.

Figure 9 caption: Provide a reference for the Mann-Kendall trend method and describe the Sen's slope estimator.

Lines 868 – 870: This sentence is confusing. What does "almost all stations have either statistically significant decreasing or not statistically significant trends in the absorption coefficient" mean? Does it mean that the only ss trends in absorption are decreasing trends?

Lines 873 – 879: It is a little frustrating that hints of interesting trends are mentioned ("Polar stations exhibit a mix of increasing and decreasing trends") without more detailed explanation. Why is scattering at ZEP, PAL, and SPO increasing on an annual basis but decreasing at BRW? If this is discussed in more detail in the companion papers that should be explicitly stated here so that the reader knows where to find further information. Also – it's not clear what an annual average represents since there may be a decreasing trend in one season and an increasing trend in another. I am thinking of sulfate in the Arctic where it is decreasing in winter/spring due to air quality regulations but could be increasing in summer due to decreasing sea ice.

Lines 888 – 890: "...simulated trends are in agreement with SARGAN derived trends suggesting significant decreases found over North America and Europe..." This sounds like models are being used to validate measurements.

Lines 892 – 894: Is this supposed to say that "...NO statistically significant AOD and sulfate trends...".

General comment: There is heavy use of Collaud Coen et al. (submitted) and Mortier et al. (submitted) in this paper. I am not sure of AMT's policies concerning citing results from papers that have not been published yet.

---

## Referee Comment (RC2) · Anonymous Referee #2 · 3 Apr 2020

Referee comment to "A global analysis of climate-relevant aerosol properties retrieved from the network of GAW near-surface observatories" by Laj et al.

Overall comment: This paper provides the full technical descriptions and overall summaries of the in-situ aerosol primary datasets (total number concentration, scattering and absorption coefficients) from a global network of near-surface aerosol monitoring stations organized and maintained by the authors. The dataset itself will play a central role in evaluating the accuracy of aerosol models used for climate simulation. The dataset demonstrated for the first time the decadal decreasing trends in surface aerosol concentration over the globe. This should be important as quantitative evidence of the

overall outcome of the pollution mitigation efforts in many countries performed in this period. It should be published in any case.

Major critical comment: I recommend the authors remove/shrink the presentations and discussion on the "comparison with models" (section 6.2.4, Figure 8) which may distract the reader's attention from the main story of this paper. Those results can be included in the companion papers (Glib et al., Morthier et al. submitted). The model results without descriptions of the underlying assumptions (i.e., details of parameterization, emission) are not very informative to me.

Minor comments: L149. "45% of the variance": Do you mean here the inter-model variance? Please be more specific.

L156. "important" here is too colloquial. Please remove or reword it.

Table 1. "Hyphen symbol" is misused as "Minus symbol" at several places in Table 1. Please correct.

L270. Define the acronym "SARGAN" here.

L358. Define "AE31" or refer to Table 2 here.

L440. Is there any specific intention to use brackets around "product"?

L531. Define "N". I suppose it means particle number concentration.

The section titles 6.2.1.~6.2.2 are missing. Please check.

---

## Author Comment (AC1) · 19 Jun 2020

**Referee comment to "A global analysis of climate-relevant aerosol properties retrieved from the network of GAW near-surface observatories" by Laj et al.**

**Response to Anonymous Referee #1**

We would like to thank Anonymous referee #1 for the very useful comments on the manuscript. You will find below our specific answers to the different points raised in the review. All modifications are noted in red in the new version of the manuscript, sent to AMT.

The results of a large effort to summarize the climate-relevant in situ aerosol properties available at all sites connected to the GAW network are presented. The growth of the global base over the past few decades is shown as well as the increase in the usage of the data over the past ten years. Values of absorption, scattering, and SSA are compared for the wide range of existing sites. Seasonal data and annual trends are also presented. All in all, it is a substantial effort that is definitely worthy of publication in AMT. Relatively minor comments are listed below.

Table 1. Should the description of CCN say "CCN number concentration is sometimes approximated using the fraction of particles larger than a given diameter from the particle number size distribution AND AN ASSUMED CHEMICAL COMPOSITION"?
In principle, this is correct. However, it is true that many studies are approximating the CCN number concentration using the fraction of particles larger than 100 nm, without any consideration on their chemical composition/hygroscopicity. This is often the case for long-term records where chemical composition is not necessarily avaialable with the required time resolution (1h). We have made this point clearer in the Table adding « neglecting the influence of particle chemical composition »

Line 225: "...but also including sites IN other WMO regions..."
This is corrected in the manuscript

Line 249: "..many of them ARE no longer documented..."
This is corrected in the manuscript

Line 271: Analyzes instead of analyses?
In the paper, we have constantly used the UK wording with « s » and not the US with « z ». So we did not implement that change.

Line 271: Define SARGAN here where it is first used.
This is corrected in the manuscript

Lines 320 – 325: It would be handy to have a reference to the relevant report of the GAW measurement guidelines included in Table 2 , i.e. WMO/GAW Report #200 for light scattering and absorption.
References to all WMO/GAW reports are actually listed in the original version of the manuscript.

Line 364: define the ultrafine and fine ranges in terms of diameter.
Ranges are now defined (10-100 nm and 100-1000 nm)

Lines 400 – 413: When was this change made, i.e. only removing data affected by instrument issues or contamination? Were older data sets amended?
The change in quality control approach was induced by the shifting objectives of WMO and thus GAW. These become visible in the shift between the GAW Strategic Plan: 2008 – 2015 and the GAW Implementation Plan 2016 – 2023. While the 2008-2015 Strategic Plan talks about "Changes in the weather and climate related to human influence" and "Risk reduction of air pollution on human health

and issues involving long-range transport and deposition of air pollution", the 2016 – 2023 GAW Implementation Plan mentions "Strengthen capabilities to predict climate, weather and air quality". This reflects GAWs shift from observations relevant for climate and long-range transport towards services and further application areas such as air quality prediction. The shift from single purpose to multi purpose quality control was thus introduced in 2016. Due to the lack of recent guidelines for GAW aerosol data quality control, this change was communicated through project meetings and bilateral contact with data providers. Due to the scientific independence of GAW stations, implementation of the new data QC approach varies. Only few data providers re-processed older data due to work load restrictions, but most have adopted the new QC policy.

To reflect this information, the text is changed to "As of 2016, it was acknowledged that atmospheric composition data serves multiple purposes and applications. This is reflected by the recommendation to only remove data affected by instrument issues or contamination during quality control, and indicate local or regional influence with a flag that leaves the data valid. This implies, for any application of WDCA data, filtering the data according to purpose is the first step. When using WDCA data, this shift in quality control approach, which may vary among stations due to their scientific independence, needs to be taken into account. Due to resource limitations, data before 2016 was mostly not reprocessed."

**Lines 439 – 440: Are there references for the "manual-expansion type" and "automated version" that can be provided so the reader knows what these particle counters are?**
Proper reference has been added in the new version of the manuscript : Hogan, A.W., and Gardner, G. (1968) A nucleus counter of increased sensitivity. J. Rech. Atmos. 3:59-61.

**Line 484: define kerbside.**
This is now defined : (near-road)

**Line 540: A couple of sentences with references on different methods that have been used to calculate trends would be helpful here.**
The following sentences were added to the section : « The non-parametric seasonal Mann-Kendall (MK) statistical test associated with several prewhitening methods and with the Sen's slope was used as main trend analysis method (Collaud Coen et al., 2020 submitted). Comparisons with General Least Mean Square associated with Autoregressive Bootstrap (GLS/ARB) and with standard Least Mean Square analysis (LMS) (Asmi et al., 2013, Collaud Coen et al., 2013) enabled confirmation of the detected MK statistically significant trends and the assessment of advantages and limitations of each method »

**Line 551: ". . ..too high, however, OBSERVATIONS INDICATE it is. . .."**
This is now corrected in the new version of the manuscript

**Lines 551 – 552: What exactly is meant by "While the bias values are robust at the sites investigated..."? The bias values (i.e. model – measurement differences) are well characterized or low?**
The reviewer is right, the sentence was not very clear. We have rephrased the whole sentence and the sentence before and it reads now: " This would indicate that the overall OA/OC ratio in the models is too high, although many model assume for primary OC emissions a low OA/OC factor of 1.4. Secondary organic aerosol formation increases this ratio in global aerosol burdens. Note that the biases established are for the relatively few remote sites investigated. It is currently difficult to assess, if there is a robust global bias in OA, OC or its ratio for the models in question."

**Lines 565 – 569: How did the measured and modelled number size distributions compare?**
We have modified the sentence to : Kok et al. (2017), showing that dust found in the atmosphere is substantially coarser than represented in current global climate models, suggest that AeroCom models do not have a sufficient coarse dust component, which suggests that dust may even have a warming direct radiative effect.

Line 570: ". . .is the evaluation OF MEASURED AND MODELLED cloud condensation C2 nuclei."
This is now corrected in the new version of the manuscript

Line 580: Provide the link to GAWSIS here where it is first mentioned.
This is now added to the manuscript

Line 587 – 589: What is the connection between not all GAW stations being able to measure all variables listed in Table 1 and SARGAN being a subset of stations in GAW? Please clarify.
This is now clarified in line 592 : SARGAN is, therefore, a subset of stations in GAW providing in-situ aerosol variables from ground-based stations.

Line 628: Where are WMO regions I, II, III, and IV? I don't think this is stated previously.
This is now provided in the text

Line 634: What is meant by "a station footprint that is large"? Is this related to its representativeness of a region? Or land type?
We have now added : (influenced by air masses transported more than 100 km away) in the text to provide a better definition of a large footprint

Line 735: ". . .for 29 of these sites IT was possible. . ."
This is now corrected

Lines 744 – 746: Is it possible to cite a reference for this assumption?
AAE=1 was chosen for the harmonization between different devices and wavelengths as suggested by Zanatta et al., (2016, https://doi.org/10.1016/j.atmosenv.2016.09.035), now added as a reference to this assumption

.

Line 772: It is more commonly thought that Cape Grim is a coastal Southern site, than a Pacific site.
This is corrected in the new version

Table 3: Abbreviations shown in the plots (DE, RB, U) should be defined in the caption.
This is now defined in Caption of Figure 6

Figure 8a: The title says monthly means while the caption says seasonal means. Given the number of points, I assume it is the former.
Figure 8 has now been changed, responding to the request by Reviewer #2. The comment does not apply anymore

Figure 8: Coloring the points by month may provide useful information on under- and overestimates by the models. Also – 8a shows absorption and 8b shows scattering but scattering is introduced in the text first.
Figure 8 has now been changed, responding to the request by Reviewer #2. The comment does not apply anymore

Figure 9 caption: Provide a reference for the Mann-Kendall trend method and describe the Sen's slope estimator.
The reference to Collaud Coen et al., submitted was added to the caption and to the reference list

Lines 868 – 870: This sentence is confusing. What does "almost all stations have either statistically significant decreasing or not statistically significant trends in the absorption coefficient" mean? Does it mean that the only ss trends in absorption are decreasing trends?

Yes, the only statistically significant trends in absorption are decreasing trends. In fact, some corrections were done for the revised manuscript (companion paper Collaud Coen et al., ACP 2020, submitted) so that no station have an annual statistically significant increasing trend in absorption any longer. The sentence is the present manuscript was modified to:" The trends of the absorption coefficient are ss decreasing or not ss for all the stations."

Lines 873 – 879: It is a little frustrating that hints of interesting trends are mentioned ("Polar stations exhibit a mix of increasing and decreasing trends") without more detailed explanation. Why is scattering at ZEP, PAL, and SPO increasing on an annual basis but decreasing at BRW? If this is discussed in more detail in the companion papers that should be explicitly stated here so that the reader knows where to find further information. Also – it's not clear what an annual average represents since there may be a decreasing trend in one season and an increasing trend in another. I am thinking of sulfate in the Arctic where it is decreasing in winter/spring due to air quality regulations but could be increasing in summer due to decreasing sea ice.

The companion paper does not solve the reasons why, contrarily to ZEP, PAL and SPO, BRW has a decreasing trend in scattering coefficient. It has to be noted that ALT also has a decreasing trend in scattering. Potential reasons are however described in the paper: " PAL, the northernmost station, has a ss positive trend. PAL is geographically situated in Europe but it can be climatologically considered an arctic station (Schmeisser et al. 2018). PAL (slope=0.06 Mm-1/y) has a similar trend as ZEP (slope=0.05 Mm.-1./y), the nearest Arctic station, with the largest ss trend in summer (JJA) when PAL is largely influenced by Arctic air masses. The increasing trend at PAL may be due to increasing biogenic secondary organic aerosol formation related to emissions from the surrounding boreal forest (Lihavainen et al., 2015a), changes in circulation patterns or a larger influence of open water with increasing concentration of sea salt aerosol."

Lines 888 – 890: "...simulated trends are in agreement with SARGAN derived trends suggesting significant decreases found over North America and Europe..." This sounds like models are being used to validate measurements.

The sentence was changed to : For both variables, simulated trends can reproduce SARGAN derived trends suggesting significant decreases found over North America and Europe, although it must be considered that the number of models providing trends in $\sigma_{ap}$ and $\sigma_{sp}$ remains limited.

Lines 892 – 894: Is this supposed to say that "...NO statistically significant AOD and sulfate trends...".

It is actually **non** statistically. This is now corrected.

General comment: There is heavy use of Collaud Coen et al. (submitted) and Mortier et al. (submitted) in this paper. I am not sure of AMT's policies concerning citing results from papers that have not been published yet.

We assume AMT/ACP policy allows for referencing submitted papers. Collaud Coen et al, is now accepted for publication in ACP and it appears that companion papers Gliβ et al, and Mortier et al., are close to being accepted having to deal with mostly minor modifications.

---

## Author Comment (AC2) · 19 Jun 2020

**Referee comment to "A global analysis of climate-relevant aerosol properties retrieved from the network of GAW near-surface observatories" by Laj et al.**

**Response to Anonymous Referee #2**

We would like to thank Anonymous referee #2 for the very useful comments on the manuscript. You will find below our specific answers to the different points raised in the review. All modifications are noted in red in the new version of the manuscript, sent to AMT.

Overall comment: This paper provides the full technical descriptions and overall sum- maries of the in-situ aerosol primary datasets (total number concentration, scattering and absorption coefficients) from a global network of near-surface aerosol monitoring stations organized and maintained by the authors. The dataset itself will play a central role in evaluating the accuracy of aerosol models used for climate simulation. The dataset demonstrated for the first time the decadal decreasing trends in surface aerosol concentration over the globe. This should be important as quantitative evidence of the overall outcome of the pollution mitigation efforts in many countries performed in this period. It should be published in any case.

Major critical comment: I recommend the authors remove/shrink the presentations and discussion on the "comparison with models" (section 6.2.4, Figure 8) which may distract the reader's attention from the main story of this paper. Those results can be included in the companion papers (Glib et al., Morthier et al. submitted). The model results without descriptions of the underlying assumptions (i.e., details of parameterization, emission) are not very informative to me.

Thanks for this comment. We have actually modified and shorten the section, and also move it to a new section 7. We believe it is however important that this section is maintained in Laj et al., as both submitted manuscripts - Gli $\beta$  et al, submitted and Mortier et al., submitted - are not only using SARGAN data but other variables from other observations from the ground and from space. We consider it is important to maintain a clear Observation/Modelling section in Laj et al., so that agreements and discrepancies for the specific case of SARGAN variables can be discussed specifically. The section is now, however, a bit shorter, and graphs have been simplified to more clearly illustrate model performances to reproduce observations. It is also important to mention that some numbers were actually updated following comments from referees in the review process of Gli $\beta$  et al, and Mortier et al. Both papers are now sent back to ACP editors.

The manuscript now reads as follow :

[revised manuscript text omitted]

---

## Author Response (AR2)

**Response to Associate Editor Comments**

**Paolo Laj 6 July 2020**

We would like to thank the associate editor for handling the review of the paper.

Please find the response to the minor comments

Associate Editor Decision: Publish subject to technical corrections (25 Jun 2020) by
Charles Brock
Comments to the Author:
Dear Paolo et al:

Thank you for your responsiveness to the reviewers' comments. There are a few
technical clean-ups to be made before publication (although I'm sure the excellent
Copernicus editorial staff would find these!):

1) Line 930, change from "almost doubled" to "more than doubled"
   *This is modified*
2) In the revised Sect. 7.2, you introduce the abbreviation "ss" for "statistically
   significant", then fail to use it in many cases.
   *There is no more use of the abbreviation now*
3) Fig. 2 appears to be a slightly blurry bitmapped image. Do you have a vector-
   based version of this image? Or a higher-resolution bitmap?
   *All figures are now provided in high-definition*
4) There are a bunch of minor technical issues in the references; might as well fix
   them now and save the technical staff the trouble:
   *All references have been rechecked and are hopefully in the right format with DOIs*
   *included, besides one reference from 1968.*

a) Several references have the article titles in quote marks.
b) At least one reference (Kahn et al.; there are probably more) have all-capitalized
article titles
c) Journals are unevenly abbreviated. Sometimes the journal is abbreviated
appropriately, and sometimes not at all.
d) The manuscripts under review should be cited as ACPD titles, with appropriate DOIs.
e) Please make sure all references have DOIs (I haven't checked for this)

These sorts of reference errors are typical of EndNote-type software; they ALWAYS need
to be checked manually, unfortunately.

**Referee comment to "A global analysis of climate-relevant aerosol properties retrieved from the network of GAW near-surface observatories" by Laj et al.**

**Response to Anonymous Referee #1**

We would like to thank Anonymous referee #1 for the very useful comments on the manuscript. You will find below our specific answers to the different points raised in the review. All modifications are noted in red in the new version of the manuscript, sent to AMT.

The results of a large effort to summarize the climate-relevant in situ aerosol properties available at all sites connected to the GAW network are presented. The growth of the global base over the past few decades is shown as well as the increase in the usage of the data over the past ten years. Values of absorption, scattering, and SSA are compared for the wide range of existing sites. Seasonal data and annual trends are also presented. All in all, it is a substantial effort that is definitely worthy of publication in AMT. Relatively minor comments are listed below.

Table 1. Should the description of CCN say "CCN number concentration is sometimes approximated using the fraction of particles larger than a given diameter from the particle number size distribution AND AN ASSUMED CHEMICAL COMPOSITION"?
In principle, this is correct. However, it is true that many studies are approximating the CCN number concentration using the fraction of particles larger than 100 nm, without any consideration on their chemical composition/hygroscopicity. This is often the case for long-term records where chemical composition is not necessarily avaialable with the required time resolution (1h). We have made this point clearer in the Table adding « neglecting the influence of particle chemical composition »

Line 225: ". . .but also including sites IN other WMO regions. . ."
This is corrected in the manuscript

Line 249: "..many of them ARE no longer documented. . ."
This is corrected in the manuscript

Line 271: Analyzes instead of analyses?
In the paper, we have constantly used the UK wording with « s » and not the US with « z ». So we did not implement that change.

Line 271: Define SARGAN here where it is first used.
This is corrected in the manuscript

Lines 320 – 325: It would be handy to have a reference to the relevant report of the GAW measurement guidelines included in Table 2 , i.e. WMO/GAW Report #200 for light scattering and absorption.
References to all WMO/GAW reports are actually listed in the original version of the manuscript.

Line 364: define the ultrafine and fine ranges in terms of diameter.
Ranges are now defined (10-100 nm and 100-1000 nm)

Lines 400 – 413: When was this change made, i.e. only removing data affected by instrument issues or contamination? Were older data sets amended?
The change in quality control approach was induced by the shifting objectives of WMO and thus GAW. These become visible in the shift between the GAW Strategic Plan: 2008 – 2015 and the GAW Implementation Plan 2016 – 2023. While the 2008-2015 Strategic Plan talks about "Changes in the weather and climate related to human influence" and "Risk reduction of air pollution on human health and issues involving long-range transport and deposition of air pollution", the 2016 – 2023 GAW

Implementation Plan mentions "Strengthen capabilities to predict climate, weather and air quality". This reflects GAWs shift from observations relevant for climate and long-range transport towards services and further application areas such as air quality prediction. The shift from single purpose to multi purpose quality control was thus introduced in 2016. Due to the lack of recent guidelines for GAW aerosol data quality control, this change was communicated through project meetings and bilateral contact with data providers. Due to the scientific independence of GAW stations, implementation of the new data QC approach varies. Only few data providers re-processed older data due to work load restrictions, but most have adopted the new QC policy.

To reflect this information, the text is changed to "As of 2016, it was acknowledged that atmospheric composition data serves multiple purposes and applications. This is reflected by the recommendation to only remove data affected by instrument issues or contamination during quality control, and indicate local or regional influence with a flag that leaves the data valid. This implies, for any application of WDCA data, filtering the data according to purpose is the first step. When using WDCA data, this shift in quality control approach, which may vary among stations due to their scientific independence, needs to be taken into account. Due to resource limitations, data before 2016 was mostly not reprocessed."

Lines 439 – 440: Are there references for the "manual-expansion type" and "automated version" that can be provided so the reader knows what these particle counters are?

Proper reference has been added in the new version of the manuscript : Hogan, A.W., and Gardner, G. (1968) A nucleus counter of increased sensitivity. J. Rech. Atmos. 3:59-61.

Line 484: define kerbside.

This is now defined : (near-road)

Line 540: A couple of sentences with references on different methods that have been used to calculate trends would be helpful here.

The following sentences were added to the section : « The non-parametric seasonal Mann-Kendall (MK) statistical test associated with several prewhitening methods and with the Sen's slope was used as main trend analysis method (Collaud Coen et al., 2020 submitted). Comparisons with General Least Mean Square associated with Autoregressive Bootstrap (GLS/ARB) and with standard Least Mean Square analysis (LMS) (Asmi et al., 2013, Collaud Coen et al., 2013) enabled confirmation of the detected MK statistically significant trends and the assessment of advantages and limitations of each method »

Line 551: ". . ..too high, however, OBSERVATIONS INDICATE it is. . .."

This is now corrected in the new version of the manuscript

Lines 551 – 552: What exactly is meant by "While the bias values are robust at the sites investigated..."? The bias values (i.e. model – measurement differences) are well characterized or low?

The reviewer is right, the sentence was not very clear. We have rephrased the whole sentence and the sentence before and it reads now:  " This would indicate that the overall OA/OC ratio in the models is too high, although many model assume for primary OC emissions a low OA/OC factor of 1.4. Secondary organic aerosol formation increases this ratio in global aerosol burdens. Note that the biases established for the relatively few remote sites investigated. It is currently difficult to assess, if there is a robust global bias in OA, OC or its ratio for the models in question."

Lines 565 – 569: How did the measured and modelled number size distributions compare?

We have modified the sentence to : Kok et al. (2017), showing that dust found in the atmosphere is substantially coarser than represented in current global climate models, suggest that AeroCom models do not have a sufficient coarse dust component, which suggests that dust may even have a warming direct radiative effect.

Line 570: "...is the evaluation OF MEASURED AND MODELLED cloud condensation C2 nuclei."
This is now corrected in the new version of the manuscript

Line 580: Provide the link to GAWSIS here where it is first mentioned.
This is now added to the manuscript

Line 587 – 589: What is the connection between not all GAW stations being able to measure all variables listed in Table 1 and SARGAN being a subset of stations in GAW? Please clarify.
This is now clarified in line 592 : SARGAN is, therefore, a subset of stations in GAW providing in-situ aerosol variables from ground-based stations.

Line 628: Where are WMO regions I, II, III, and IV? I don't think this is stated previously.
This is now provided in the text

Line 634: What is meant by "a station footprint that is large"? Is this related to its representativeness of a region? Or land type?
We have now added : (influenced by air masses transported more than 100 km away) in the text to provide a better definition of a large footprint

Line 735: "...for 29 of these sites IT was possible..."
This is now corrected

Lines 744 – 746: Is it possible to cite a reference for this assumption?
AAE=1 was chosen for the harmonization between different devices and wavelengths as suggested by Zanatta et al., (2016, https://doi.org/10.1016/j.atmosenv.2016.09.035), now added as a reference to this assumption

.

Line 772: It is more commonly thought that Cape Grim is a coastal Southern site, than a Pacific site.
This is corrected in the new version

Table 3: Abbreviations shown in the plots (DE, RB, U) should be defined in the caption.
This is now defined in Caption of Figure 6

Figure 8a: The title says monthly means while the caption says seasonal means. Given the number of points, I assume it is the former.
Figure 8 has now been changed, responding to the request by Reviewer #2. The comment does not apply anymore

Figure 8: Coloring the points by month may provide useful information on under- and overestimates by the models. Also – 8a shows absorption and 8b shows scattering but scattering is introduced in the text first.
Figure 8 has now been changed, responding to the request by Reviewer #2. The comment does not apply anymore

Figure 9 caption: Provide a reference for the Mann-Kendall trend method and describe the Sen's slope estimator.
The reference to Collaud Coen et al., submitted was added to the caption  and to the reference list

Lines 868 – 870: This sentence is confusing. What does "almost all stations have either statistically significant decreasing or not statistically significant trends in the absorption coefficient" mean? Does it mean that the only ss trends in absorption are decreasing trends?

Yes, the only statistically significant trends in absorption are decreasing trends. In fact, some corrections were done for the revised manuscript (companion paper Collaud Coen et al., ACP 2020, submitted) so that no station have an annual statistically significant increasing trend in absorption any longer. The sentence is the present manuscript was modified to:" The trends of the absorption coefficient are ss decreasing or not ss for all the stations."

Lines 873 – 879: It is a little frustrating that hints of interesting trends are mentioned ("Polar stations exhibit a mix of increasing and decreasing trends") without more detailed explanation. Why is scattering at ZEP, PAL, and SPO increasing on an annual basis but decreasing at BRW? If this is discussed in more detail in the companion papers that should be explicitly stated here so that the reader knows where to find further information. Also – it's not clear what an annual average represents since there may be a decreasing trend in one season and an increasing trend in another. I am thinking of sulfate in the Arctic where it is decreasing in winter/spring due to air quality regulations but could be increasing in summer due to decreasing sea ice.

The companion paper does not solve the reasons why, contrarily to ZEP, PAL and SPO, BRW has a decreasing trend in scattering coefficient. It has to be noted that ALT also has a decreasing trend in scattering. Potential reasons are however described in the paper: " PAL, the northernmost station, has a ss positive trend. PAL is geographically situated in Europe but it can be climatologically considered an arctic station (Schmeisser et al. 2018). PAL (slope=0.06 Mm-1/y) has a similar trend as ZEP (slope=0.05 Mm.-1./y), the nearest Arctic station, with the largest ss trend in summer (JJA) when PAL is largely influenced by Arctic air masses. The increasing trend at PAL may be due to increasing biogenic secondary organic aerosol formation related to emissions from the surrounding boreal forest (Lihavainen et al., 2015a), changes in circulation patterns or a larger influence of open water with increasing concentration of sea salt aerosol."

Lines 888 – 890: "...simulated trends are in agreement with SARGAN derived trends suggesting significant decreases found over North America and Europe..." This sounds like models are being used to validate measurements.

The sentence was changed to : For both variables, simulated trends can reproduce SARGAN derived trends suggesting significant decreases found over North America and Europe, although it must be considered that the number of models providing trends in $\sigma_{ap}$ and $\sigma_{sp}$ remains limited.

Lines 892 – 894: Is this supposed to say that "...NO statistically significant AOD and sulfate trends...".

It is actually **non** statistically. This is now corrected.

General comment: There is heavy use of Collaud Coen et al. (submitted) and Mortier et al. (submitted) in this paper. I am not sure of AMT's policies concerning citing results from papers that have not been published yet.

We assume AMT/ACP policy allows for referencing submitted papers. Collaud Coen et al, is now accepted for publication in ACP and it appears that companion papers Gliβ et al, and Mortier et al., are close to being accepted having to deal with mostly minor modifications.

**Referee comment to "A global analysis of climate-relevant aerosol properties retrieved from the network of GAW near-surface observatories" by Laj et al.**

**Response to Anonymous Referee #2**

We would like to thank Anonymous referee #2 for the very useful comments on the manuscript. You will find below our specific answers to the different points raised in the review. All modifications are noted in red in the new version of the manuscript, sent to AMT.

Overall comment: This paper provides the full technical descriptions and overall sum- maries of the in-situ aerosol primary datasets (total number concentration, scattering and absorption coefficients) from a global network of near-surface aerosol monitoring stations organized and maintained by the authors. The dataset itself will play a central role in evaluating the accuracy of aerosol models used for climate simulation. The dataset demonstrated for the first time the decadal decreasing trends in surface aerosol concentration over the globe. This should be important as quantitative evidence of the overall outcome of the pollution mitigation efforts in many countries performed in this period. It should be published in any case.

Major critical comment: I recommend the authors remove/shrink the presentations and discussion on the "comparison with models" (section 6.2.4, Figure 8) which may distract the reader's attention from the main story of this paper. Those results can be included in the companion papers (Glib et al., Morthier et al. submitted). The model results without descriptions of the underlying assumptions (i.e., details of parameterization, emission) are not very informative to me.

Thanks for this comment. We have actually modified and shorten the section, and also move it to a new section 7. We believe it is however important that this section is maintained in Laj et al., as both submitted manuscripts - Gliβ et al, submitted and Mortier et al., submitted - are not only using SARGAN data but other variables from other observations from the ground and from space. We consider it is important to maintain a clear Observation/Modelling section in Laj et al., so that agreements and discrepancies for the specific case of SARGAN variables can be discussed specifically. The section is now, however, a bit shorter, and graphs have been simplified to more clearly illustrate model performances to reproduce observations. It is also important to mention that some numbers were actually updated following comments from referees in the review process of Gliβ et al, and Mortier et al. Both papers are now sent back to ACP editors.

The manuscript now reads as follow :

[revised manuscript text omitted]

Minor comments: L149. "45% of the variance": Do you mean here the inter-model variance? Please be more specific.

This refers to Carslaw et al., paper in 2013. It refers indeed to the variance of a multi-model ensemble when varying the reference year for the pre-anthropogenic emissions. This is now added to the new version.

L156. "important" here is too colloquial. Please remove or reword it.

« important » is now removed

Table 1. "Hyphen symbol" is misused as "Minus symbol" at several places in Table 1. Please correct.

We have checked and it seems that we are using the « Minus symbol » and not the « Hyphen symbol » throughout the Table. This could be rechecked during the editing process.

L270. Define the acronym "SARGAN" here.

This is now defined when first used

L358. Define "AE31" or refer to Table 2 here.

This is now corrected

L440. Is there any specific intention to use brackets around "product"?

The proper reference has been added to explain the use of brackets

L531. Define "N". I suppose it means particle number concentration.

N is now defined when first use

The section titles 6.2.1.~6.2.2 are missing. Please check.

This is now corrected